# Holographic tomographic volumetric additive manufacturing

Maria Isabel Álvarez-Castaño [1], Andreas Gejl Madsen[2], Jorge Madrid-Wolff [1,3], Viola Sgarminato [1], Antoine Boniface[1,4], Jesper Glückstad [2] & Christophe Moser [1]

Several 3D light-based printing technologies have been developed that rely on the photopolymerization of liquid resins. A recent method, so-called Tomographic Volumetric Additive Manufacturing, allows the fabrication of micro-scale objects within tens of seconds without the need for support structures. This method works by projecting intensity patterns, computed via a reverse tomography algorithm, into a photocurable resin from different angles to produce a desired 3D shape when the resin reaches the polymerization threshold. Printing using incoherent light patterning has been previously demonstrated. In this work, we show that a light engine with holographic phase modulation unlocks new potential for volumetric printing. The light projection efficiency is improved by at least a factor 20 over amplitude coding with diffraction-limited resolution and its flexibility allows precise light control across the entire printing volume. We show that computer-generated holograms implemented with tiled holograms and point-spread-function shaping mitigates the speckle noise which enables the fabrication of millimetric 3D objects exhibiting negative features of 31 $\mu m$ in less than a minute with a 40 mW light source in acrylates and scattering materials, such as soft cell-laden hydrogels, with a concentration of 0.5 million cells per mL.

Fabricating objects with complex geometry has become much simpler thanks to additive manufacturing. Light-based 3D printing exploits the ability of certain light-sensitive molecules to trigger polymerization or crosslinking reactions in liquid resins, thus solidifying them. Photopolymerization can be produced sequentially, as in stereolithography[1] or digital light processing[2,3], and more recently also volumetrically, as in two-photon polymerization[4,5], light-sheet microprinting[6], holographic multi-beam interference[7], or tomographic volumetric additive manufacturing[8,9]. By printing in a volumetric fashion, support struts are no longer needed, enabling the fabrication of designs with cavities and overhangs.

In Tomographic Volumetric Additive Manufacturing (TVAM), an entire three-dimensional object is simultaneously solidified by irradiating a volume of photocurable liquid resin from multiple angles with dynamic light patterns until the photocurable resin reaches a polymerization dose threshold[8,10]. Unlike most other additive manufacturing methods, TVAM is layer-less, meaning that it does not fabricate objects by solidifying one voxel, line, or layer at a time. Instead, light from sequential tomographic patterns builds up an energy dose that approximates the volume of the target object. Typical printing times are tens of seconds[11] for cm-scale prints with resolutions down to 50–80 $\mu m$[9,12] using high-power laser diodes[9,13,14]. Additionally, the technique has proven versatile and has been used to fabricate objects in materials such as acrylates, thiol-enes[12,15–17], nanoparticle-loaded composites[12], polymer-derived ceramics[18], epoxies[19], silk bioinks[20], and cell-laden hydrogels[10,21,22].

[1]Laboratory of Applied Photonics Devices, School of Engineering, Ecole Polytechnique Fédérale de Lausanne, Lausanne, Switzerland. [2]SDU Centre for Photonics Engineering, University of Southern Denmark, Odense M, Denmark. [3]Present address: Readily3D, EPFL Innovation Park, Bât. A, Lausanne, Switzerland. [4]Present address: AMS Osram, Martigny, Switzerland. ✉e-mail: maria.alvarezcastano@epfl.ch; christophe.moser@epfl.ch

Implementations of tomographic VAM rely on the Radon transform (ray optics) to compute the light amplitude patterns. Essentially, different techniques have been used to improve the light dose delivered to the photoresin. Due to the positive constraint applied to the intensity patterns after the Filtered Back Projection (FBP), the computed projection patterns produce a 3D dose with artifacts[22,23]. The dose threshold required to solidify the photosensitive resin provides an additional degree of freedom for improved shape fidelity. Iterative optimization schemes that use appropriate error functions to minimize printed voxel errors have proven to be very effective[8,24,25]. The diffusion of inhibiting molecules has been modeled as a convolution operation and shown to improve fidelity[26]. Additionally, including 3D imaging methods during fabrication can be used to experimentally correct for oxygen diffusion or other effects such as temperature-induced reaction kinetics[9,26–28]. The Radon transform assumes straight rays to compute the projected light patterns. Projection patterns can be thought of as "extruded" two-dimensional patterns that propagate as a collimated beam. In practice, this is implemented by imaging a two-dimensional spatial light modulator into a cylindrical vial containing the photoresin and relying on the depth of focus of the imaging system to maintain a quasi-collimated image. If the vial's diameter is much larger than the depth of focus, blurring artifacts appear and compromise the resolution[9,14]. The space-bandwidth product of the light source is a key optical parameter, as it determines the depth of focus. A diffraction-limited spot has the lowest space bandwidth product. Additionally, in this type of optical imaging configuration, the two-dimensional spatial light modulator modulates only the intensity, not the phase. Typically, digital micromirror devices (DMD) are used as reflective binary amplitude modulators in light-based 3D printers including TVAM. Here, one pixel on the 2D modulator corresponds to one pixel in the relayed image.

In this work, we propose a light engine that harnesses the phase properties of the light beam. We term this technique HoloVAM. Phase encoding of tomographic projections offers multiple advantages over amplitude modulation. First, phase-encoding improves light efficiency, as all pixels of the display can contribute to the projected intensity pattern. Since tomographic projection patterns typically exhibit high spatial frequency information, most pixels are dark. Amplitude encoding is therefore highly inefficient: typically, less than 1% of the incident light reaches the vial. Second, phase-encoding allows a modification of the point spread function (PSF) to be encoded within the same hologram, allowing 3D digital control of the light beam, for example, by increasing the depth of focus using an Axicon phase to generate a Bessel beam, or by multi-plane projections of the same pattern by adding phases of Fresnel lenses, effectively creating a low-divergence projection. A self-healing beam, generated using a Helical Phase Plate (HPP) to produce an Optical Vortex (OV), enables printing within scattering materials. Herein, we use a projection system that converts a two-dimensional phase modulator to a two-dimensional intensity pattern in the Fourier plane. All pixels on the modulator contribute, by interference, to all pixels on the image plane. There are multiple iterative algorithms or machine learning approaches[29] that can generate two-dimensional input phase patterns to produce a user-specified intensity distribution in a desired plane or volume. The Gerchberg–Saxton (GS) iterative algorithm is one of the many methods to calculate a two-dimensional input phase pattern, which results in a desired intensity distribution pattern at the optical Fourier plane[30]. Binary holograms computed with the Gerchberg-Saxton algorithm and encoded with Lee holograms have recently been used to control multiple foci in two-photon lithography, thereby increasing printing speed[31].

In our approach, we demonstrate the use of phase encoding in tomographic VAM by printing millimeter-scale objects in an acrylate-based resin and in a hydrogel based on Gelatin Metacryloyl (GelMA) loaded with cells at a concentration of 0.5 million cells per mL. We demonstrate a printing time of less than 60 s with a single continuous-wave 40 mW laser diode at 405 nm (coupled with an efficiency of 33% in a single mode fiber) and reach negative features (holes) of 31.35 μm reliably. This short printing time is the result of the efficiency of the light engine. First, we show a comparison between the light efficiency for amplitude and phase encoding using a DMD as a spatial light modulator (SLM)[32,33]. We then present a pipeline to compute holograms for tomographic projections, where tiled holograms with PSF modification reduce speckle noise and increase contrast, and the Lee Hologram method allows to use of the DMD as a fast phase modulator[32–36].

## Results

### Optical configuration and light engine efficiency

The optical configuration of the holographic VAM is presented in Fig. 1. Unlike conventional TVAM[8,9], the DMD is arranged in a Fourier configuration, which allows the reconstruction of the projected holograms into the printing volume. We combine time-multiplexing[36] and tiled holograms[30,37] using the HoloTile method[38,39] to produce near speckle-free projections, as explained in the "method" section. A single (spatial)-mode laser at $\lambda = 405$ nm is used to achieve maximum spatial resolution in HoloVAM. A single-mode laser has less optical power than a multimode laser and thus power efficiency of the light engine is of high importance to obtain enough intensity at the build volume, keeping printing times short (~60 s). We first investigate the light efficiency difference between amplitude and phase encoding on a binary amplitude DMD spatial light modulator.

In VAM systems, due to the nature of the sparse intensity of the tomographic projections, most of the optical power incident on the 2D modulator is lost, with only a few pixels contributing to the projected image. This is exemplified in the histograms in Fig. 2b, where each pattern shows a large count of dark pixels. We estimate the light efficiency of each pattern using the following relation:

$$\eta_{patt} = \frac{\sum_{g=0}^{g_{max}} n_g \frac{g}{g_{max}}}{N_{pixels}} * \eta_{DMD} \qquad (1)$$

where $n_g$ is the number of pixels in the image that have a gray level equal to $g$, and $g_{max}$ is the maximum gray level value, where 8-bit images are used for amplitude tomographic projections, therefore $g_{max} = 255$. $N_{pixels}$ is the number of pixels in the image, and $\eta_{DMD}$ is the pixel reflectivity of the DMD at the operating wavelength of 405 nm. As an example, the histogram using theoretical projections images of the Benchy boat at projection angles $\theta = 0°$ and $\theta = 90°$ shows a light efficiency $\eta_{patt}$ for amplitude modulation of 0.36 % and 0.42 % respectively (pixel reflectivity $\eta_{DMD} = 65\%$ at 405 nm). Experimentally, the light efficiency has been measured slightly below the theoretical value (0.34 % and 0.40 % respectively). Because of this low efficiency, the VAM system from Loterie et al.[9] combines six high-power broad-area laser diodes with a nominal power of 6.4 W. The reason for this low light efficiency is that there is a one-to-one relation between the plane of the modulator and the plane of the image. In contrast to the above intensity-based projection, a holographic phase modulation with a computer-generated hologram (CGH) pattern has each pixel of the modulator contributing to the intensity pattern which yields a substantially higher light efficiency. Figure 2c shows the same reconstructed projection of the Benchy boat with a 28.6X improvement in light efficiency (see light efficiency measurements in the Supplementary Note 3).

### Holographic encoding

Speckle is inherent in holographic projection[40,41]. While a CGH can reconstruct the desired intensity distribution at the Fourier plane, the reconstruction quality is affected by the CGH design, the pixelated structure and type of spatial light modulator (phase-only or

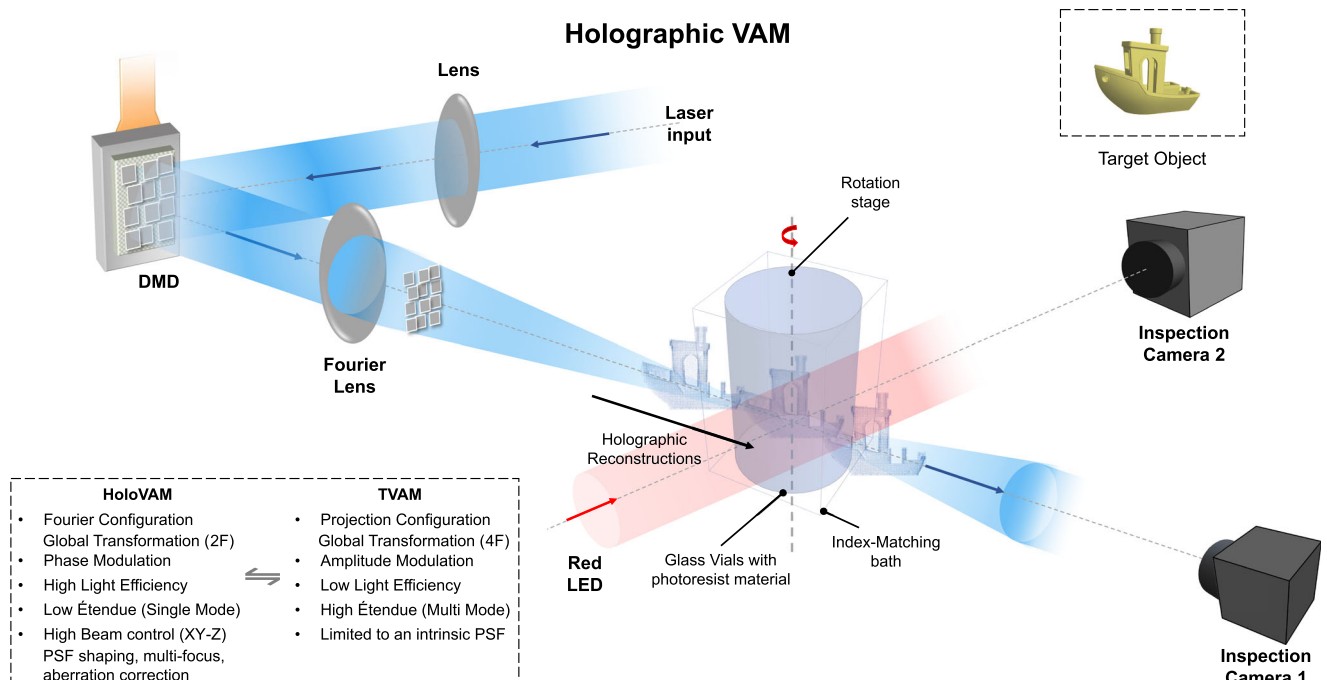

**Fig. 1 | Optical configuration for holographic volumetric additive manufacturing.** A single spatial mode laser diode at 405 nm is collimated and expanded to fit the active area of a DMD. A Fourier lens reconstructs the hologram at its Fourier plane which is located within the rotary photoresin container (more details in Supplementary Fig. 1). The holographic projections are displayed synchronously with the rotation stage. Two inspection cameras are used to monitor the holographic pattern reconstruction camera, and the polymerization process camera 2 (See Supplementary Fig. 1). Benchy boat (Copyright CC) render produced with Wolfram Mathematica® 13.1[63].

intensity-only), spurious unmodulated reflection, and optical aberrations[35,42]. A typical intensity reconstruction from a CGH displayed on a pixelated 2D display is shown in Supplementary Fig. 2a–d), which exhibits speckle noise. To improve the reconstruction quality of phase-only holograms, a plurality of methods has been proposed such as using time averaging[36,43] and iterative algorithms applying bandwidth constraints[44–46].

Spatial and/or temporal tiling of holograms can be used to mitigate the effect of unwanted crosstalk between pixels[37,38,42,47]. For this work, we use the HoloTile modality[38,39], described in detail in Supplementary Note 7, which consists of superimposing a first smaller hologram of the desired projection pattern which is spatially tiled on the available full area of the spatial light modulator and a second hologram −called here a PSF hologram which defines the shape of each reconstructed pixel in the image plane. The key aspect of tiling holograms is to provide high-speed computation, and separate reconstructed image points (modifying the original grid of image points by changing the size of the tiles). This technique essentially eliminates the uncontrolled interference at the reconstruction plane that gives rise to speckles[37,38,42].

The number of tiles $N_t$ determines the dimensions of the sub-holograms $m \times m = \frac{L}{N_t} \times \frac{L}{N_t}$. Where $L \times L$ is the dimension of a full hologram size (the full size of the SLM display, in pixels). The sub-hologram $\varphi_{SubH}(x, y)$ is calculated with the GS algorithm (see Supplementary Note 6). The sub-hologram is tiled in a $L \times L$ square hologram $\varphi_{tile}(x, y)$, which is a mosaic of $N_t^2$ sub-holograms $h(x, y)$. A PSF hologram of size $L \times L$ multiplies the tiled hologram $h(x, y)$, which is then the final hologram displayed on the SLM. To experimentally investigate the effect of tiling the holograms on the reconstruction quality, six CGHs with the same target ("A" letter) but tiled with different $N_t$ were generated using a flat-top PSF that produces a square pixel in the reconstruction plane (See Supplementary Note 7). The results of the reconstructions in the Fourier plane of a lens are shown in Fig. 3b. The reconstruction quality of the holograms compared to the desired pattern in the image plane (Fourier plane) is quantified with the following metrics: Mean Squared Error (MSE), and Peak Signal-to-Noise Ratio (PSNR). The MSE decreases and PSNR increases with the number of tiles Fig. 3c–e. A low MSE and increasing PSNR indicate a low level of speckle noise. We define the contrast over the region defined by the letter "A" as a "goodness" metric for the fidelity of the reconstruction. The higher the contrast, the better the fidelity. As the number of tiles increases, the contrast of the reconstruction increases (Fig. 3d), while the resolution decreases. In summary, there is a trade-off between speckle reduction and feature size. Since time multiplexing is applied in our approach, an initial evaluation was performed by time multiplexing six tiled holograms corresponding to $N_t = 3$–12 tiles. Figure 3b right shows the image recorded over an integration time corresponding to the projection time of the six holograms (50 ms each). Figure 3c, d shows that there is a clear trend towards increasing the PSNR and decreasing MSE when the number of tiles increases. This is due to the speckle noise reduction and smaller errors of the holographic projections with increasing number of tiles, with additional enhancement through time multiplexing.

The number of tiles and the sampling rate $\Delta x$ provide a relation to the number of controllable grid points in the reconstruction plane and the feature size respectively[37,38,42,48]. The CGH is encoded as a binary amplitude hologram for experiments on the DMD modulator using the Lee method[32,33]. Because of the use of a flat-top PSF, the projection's minimal feature size is the distance between the grid in the reconstruction plane (Fig. 3f). We improve the speckle noise at the cost of an increased feature size. For example, with 3 tiles, the feature size is 23.09 μm. Smaller feature sizes can be realized with different PSF to make use of the space between grid points in the reconstructed plane.

## Computation of holograms for tomographic projection

Figure 4 shows the pipeline to compute holographic projections for three different PSF: flat top (square output pixel), Bessel, and an Optical Vortex (OV), Fig. 4a, b, and c respectively (for PSF phase

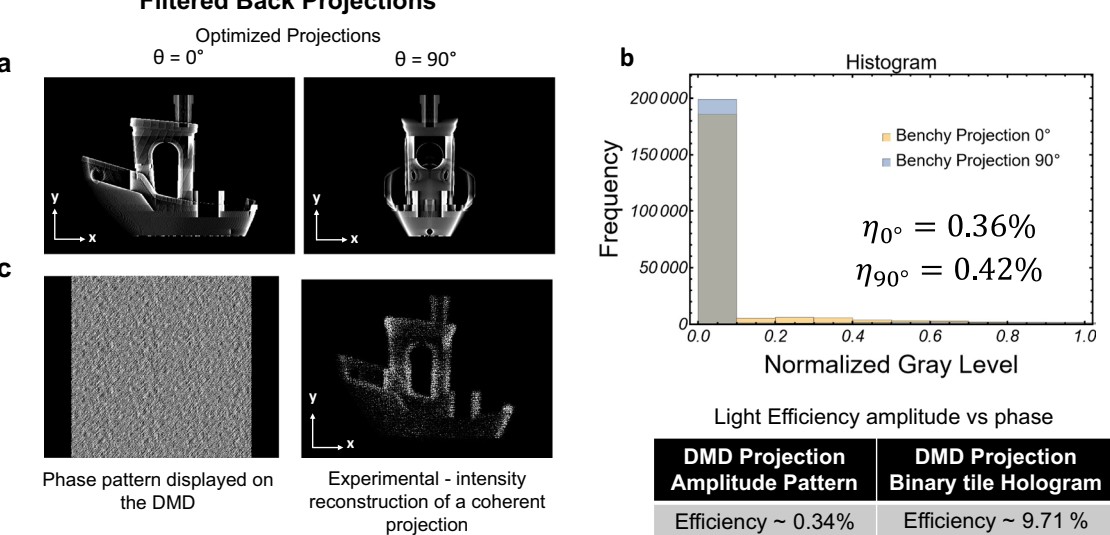

**Fig. 2 | Light efficiency: amplitude versus phase. a** Filtered back projections after applying a Ram-Lak filter and positivity constraint. **b** Histograms of two sparse projection patterns at 0 and 90 degrees. **c** (Left) Corresponding phase pattern (hologram) generated by the HoloTile technique, encoded as a Lee hologram to be able to use the DMD as a fast phase modulator. (Right) Reconstructed projection of the Benchy boat from a coherent projection (hologram). The table shows the comparison between the light intensity at the sample plane for amplitude and phase coding.

modification see Supplementary Note 7, 8). Two approaches were performed to achieve collimation of the projections. For the point spread function (PSF) generating square pixels, we extend the collimation range at each projection angle by sequentially projecting patterns at different depths $z$ within the vial using time multiplexing. This is achieved by adding a Fresnel lens, for time-varying the reconstruction plane, to the tiled computer-generated hologram (CGH), which effectively "smears" the reconstruction of the projection over the axial direction, providing precise axial control. In the second approach, a Bessel beam PSF generated with an Axicon phase on the DMD[49] (see Supplementary Note 8) provides longer collimation of the projected image than when using a flat-top type PSF. Additionally, an Optical Vortex (OV) generated by a Helical Phase Plate (HPP) produces a low divergence beam with a donut intensity distribution that carries orbital angular momentum and has self-healing properties.

The final step in this process involves encoding the phase on a digital micromirror device (DMD) using the Lee method. Illustratively, a numerical reconstructed image for one projection is shown in Fig. 4a–c (bottom left) for a CGH with a number of tiles $N_t = 4$. The corresponding Lee hologram reconstruction of the same tiled hologram is shown in Fig. 4a–c bottom right for three distinct PSF shaping. For the case of the Bessel and vortex PSF, the depth of field is longer due to its "non-diffracting" characteristic, and thus there is no need to multiplex the reconstructed image at different depths.

### Low étendue and multiplane reconstruction

With the aim of using a light engine based on phase encoding for TVAM, it is necessary to generate a beam with a low divergence that satisfies the collimation assumption of the Radon transform used to compute intensity patterns targets of the holographic projections (Supplementary Note 8). The parameter that defines spatial resolution is the etendue[9]. In this case, maintaining a collimated beam over the build volume produces a high-resolution printed part. We exploit PSF engineering to effectively reduce the divergence of the projected patterns. As mentioned, we used two approaches: first, we digitally modify the beam reconstruction using a sequence of Fresnel Lenses; second, a non-diffractive source, such as a Bessel beam, produces a beam with low etendue (Supplementary Note 8). Figure 5 shows

simulations and experiments of a Flat-top PSF propagating over 10 $mm$. With a flat-top PSF, we sequentially display multiple CGHs at different depths along the optical axis, as shown in Fig. 5a. In the experiment, 25 CGHs were projected at a rate of 8 $kHz$. The incoherent sum of the projected patterns is experimentally captured by a camera on a motorized stage that moves along the depth step as the sequence of holograms is projected by the DMD (Fig. 5b). This effectively measures the optical dose that the photoresin would receive at a given angle. The propagation distance $z = 10$ $mm$ is large compared to the intended object's footprint (~2.5 $mm$). We see that simulation and experiment are in good agreement, Fig. 5c. A simulation with a Bessel beam and Optical Vortex PSF is shown in Supplementary Note 8. A single CGH using 4 tiles was used. This simulation showed good collimation of the projected pattern over 10 $mm$.

### 3D printed objects

We experimentally demonstrated the performance of our technique by printing in two different materials, an organic acrylate-based resin, and a cell-laden hydrogel. We used the CGHs described above to fabricate millimeter-scale objects. A commercial polyacrylate resin with Diphenyl (2,4,6-trimethylbenzoyl) phosphine oxide (TPO) as a photoinitiator was used as the photoresin. Parts were printed in less than 60 s. To increase the printability of small features, TEMPO (2,2,6,6-tetramethylpiperidin-1-yl)oxidanyl, a mediator in radical polymerization, was added to the photoresist[50]. Because of its radical scavenging behavior, the addition of TEMPO increases the amount of printing time required per sample. Figure 6 shows the 3D printed results of a Benchy boat using holographic projections with different PSF shapes and different number of tiles. Pictures of the printed boat show that intricate high-resolution details, such as the sharp bow of the boat, the hollow cabin, or the chimney can be well resolved.

The pictures in Fig. 6a, b show objects printed with a Flat-top PSF, produced by time multiplexing 7 holograms (Fresnel lenses) per angle, with 3 and 4 tiles respectively. The printed parts displayed in Fig. 6c, d were achieved by projecting a single Bessel hologram per angle and using 3 and 4 tiles respectively. The 150 s printing time of the samples obtained when using TEMPO is still within the printing time of conventional tomographic VAM.

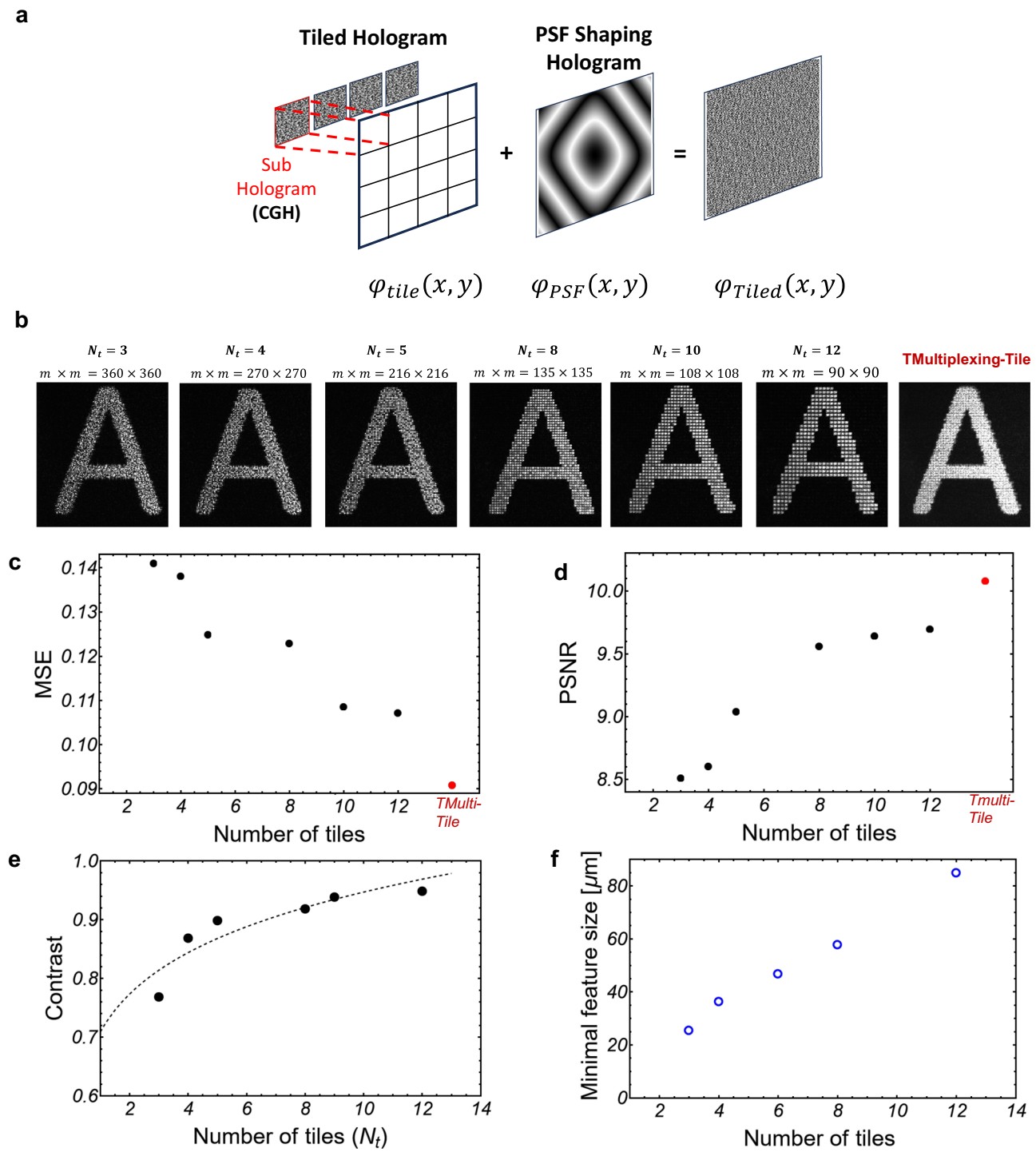

**Fig. 3 | Hologram tiling reduces speckle noise. a** Schematic of the HoloTile hologram process, where a sub-hologram is spatially tiled and convolved with a Flat-top PSF. **b** Experimental reconstruction of the tiled holograms generated of the letter "A" for a different number of tiles N$_t$. The experiments were performed using a liquid crystal SLM (See Supplementary Information Note S2, Supplementary Fig. 3). **c** Comparison of the measured mean square error (MSE), and **d** Peak signal to noise ratio (PSNR) for the different tiled holograms. **e** Reconstruction analysis considering the contrast measurements as $(I_{max} - I_{min}/I_{max} + I_{min})$. **f** Minimal projected feature size measured for different tiled holograms using the DMD. Experimental reconstruction using the DMD are shown in Supplementary Note 2.1). Supplementary Data File provides raw data for the graphs.

A well-defined hull with holes and a cylindrical chimney with a hole can be seen in the printed 3DBenchy boats in Fig. 6. The diameter of the hole in the hull was targeted at 161 $\mu m$. The prints with $N_t = 3$ (Fig. 3a, c) show a hole diameter of 112 $\mu m$ and 170 $\mu m$. The prints with $N_t = 4$ (Fig. 3b, d) show a hole diameter of 118 $\mu m$ and 140 $\mu m$. As the measurements in Fig. 3. show, the spatial feature size of the projections using $N_t = 3$ and $N_t = 4$ is expected to be 25.3 $\mu m$ and 35 $\mu m$ respectively, well within the resolution of the hull diameter. The variability of the printed hull diameter is attributed to overprinting and to the post-processing of the parts.

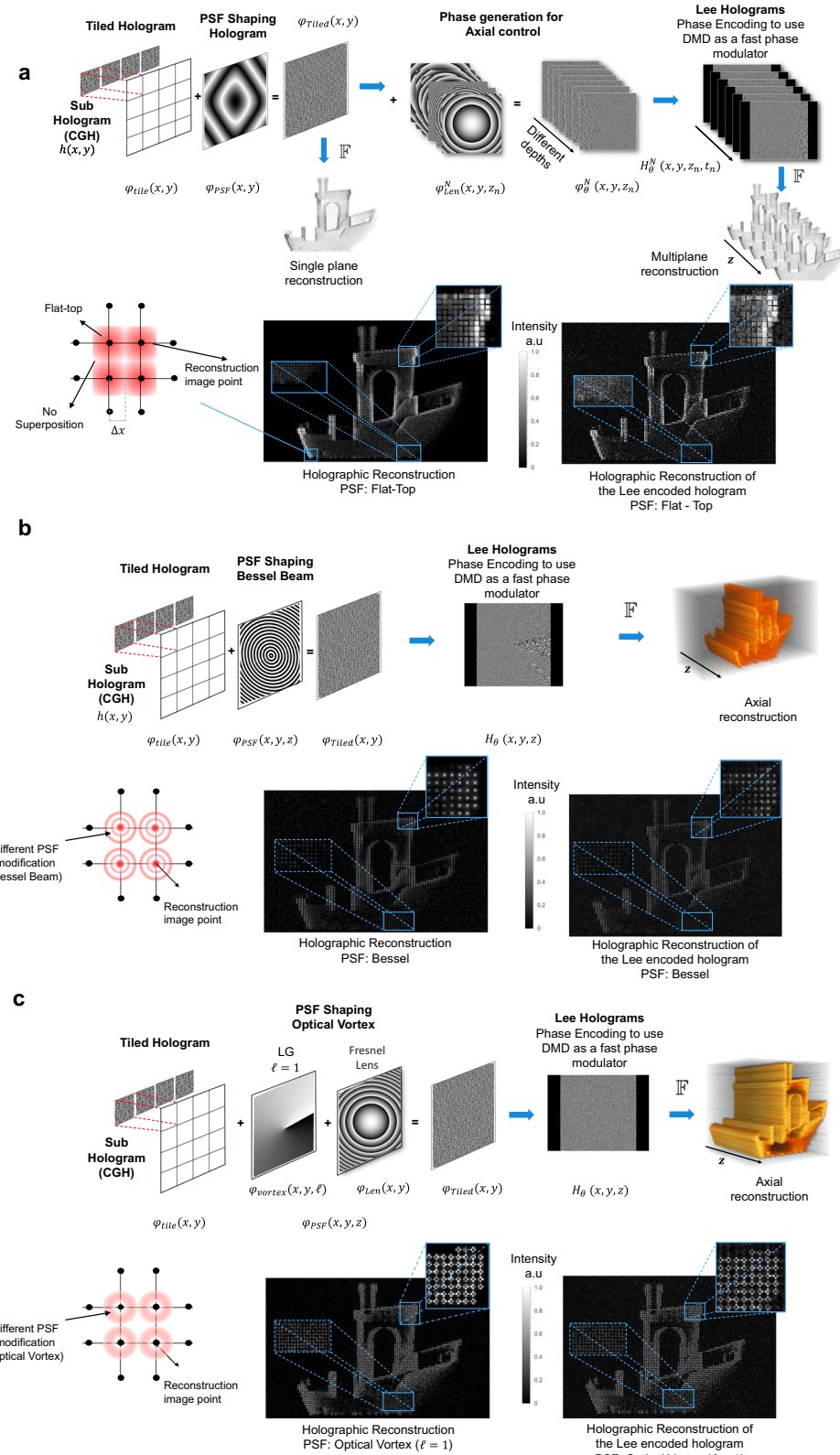

The small positive features such as the cargo box in the back deck, where the thickness is 36 $\mu m$ could not be printed with $N_t = 4$ tiles as it is too close to the minimal feature size (36.2 $\mu m$) but could be printed with $N_t = 3$ tiles (25.3 $\mu m$).

For these experiments, we measured and corrected the wobbling of the sample holder by adding a linear phase to the CGHs

(Supplementary Note 10). This correction improved the straight columns of the cabinet and chimney.

Striations are observed similarly to conventional TVAM[24], but here the stria period changes with the number of tiles, matching the minimal feature size possible with each tile. For example, with a number of tiles $N_t = 3$, the distance between grid points is 23.09 $\mu m$, close to the

**Fig. 4 | Hologram synthesis. a** Pipeline of the hologram synthesis for holographic projection for a Flat-Top. Image points illustration with a PSF modification corresponding to a flat top is shown in the bottom left. A tiled hologram reconstruction for a single projection with a flat top is illustrated in the center. Lee hologram reconstruction of the tiled phase is illustrated on the right. **b** Pipeline of the hologram synthesis for holographic projection for a Bessel PSF. Image points illustration with a PSF modification corresponding to a Bessel is shown in the bottom left. Tiled hologram reconstruction for a single projection with a Bessel is illustrated in the center. Lee hologram reconstruction of the tiled phase is illustrated on the right. **c** Hologram synthesis pipeline for the holographic projection of a vortex beam (LG, Laguerre-Gauss beam) with topological charge $\ell = 1$ using a Helical Phase Plate (see Supplementary Note 8) to generate a PSF with a donut-shape intensity. The bottom left shows an illustration of pixels with a PSF modification corresponding to an optical vortex. The tiled hologram reconstruction for a single projection using an optical vortex with charge $\ell = 1$ is shown in the middle. The Lee hologram reconstruction of the tiled phase is shown on the right.

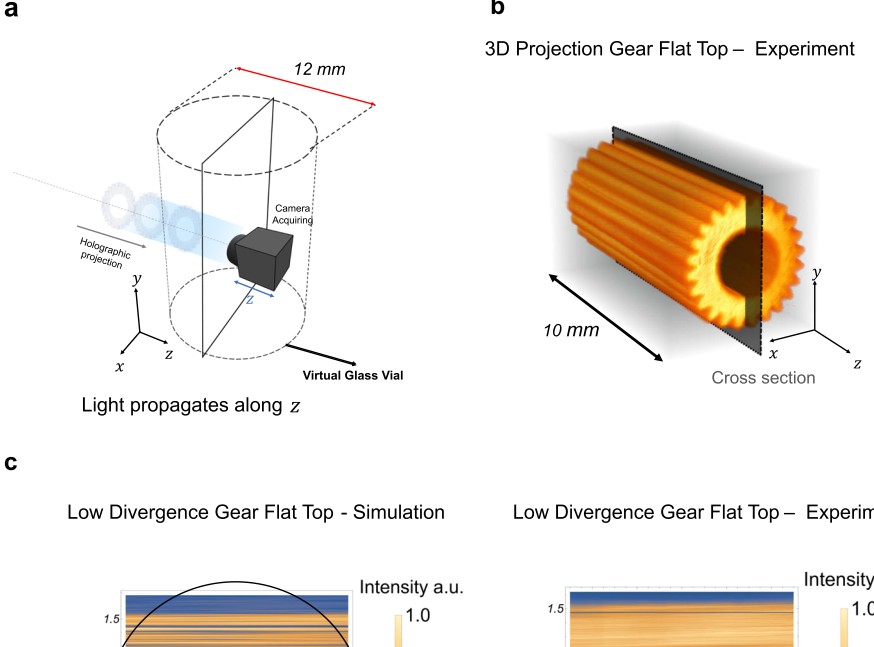

**Fig. 5 | Collimated projections thanks to multi-depth reconstruction. a** Building volume within the glass vial of 12 mm diameter. **b** Measured cumulated light projected by a sequence of 25 holograms displayed on a DMD using a flat top. **c** Right. Axial cross-section of the cumulative light dose shown in **b**. Left, simulations matching well experiment. The diameter of the glass vial is shown for size reference. Simulations using Bessel PSF and Vortex PSF are shown in Supplementary Note 8.

observed striation period of the 20 $\mu m$. Similarly, with $N_t = 4$, the distance between grid points is 36.02 $\mu m$, matching the measured 35 $\mu m$ striation period. And for $N_t = 6$, the stria period of 49 $\mu m$ and the minimal feature size is 45.5 $\mu m$, as shown in Fig. 6a,

We hypothesize that the bright centers of the flat-top convolved tiles produce the striation, mainly through a self-focusing effect[51,52]. Striation is particularly detrimental to print quality because it produces rough surfaces and objects appear white when they should appear mostly transparent. Stria reduction was quantified experimentally by modifying the point spread function of the CGH and adding time multiplexing (see Supplementary Fig. 8.4). We show that it can drastically reduce stria (Fig. 7b). The parts printed with PSFs other than the $N_t = 3$ Flat-top exhibit a smoother, shinier surface and are more transparent. We used microCT scans of the printed parts to obtain high-resolution images of their surface as shown in Fig. 7c. The inlets illustrate the side of the cabin of the boats, a region where striation occurs. The single $N_t = 3$ Flat-top PSF resulted in a rougher print. We measured the depth and the pitch of striation at the hull, the

cabin, and the chimney from these microCT scans, and observed that tiling two vortices yielded prints with significantly shallower and smaller striation than using a single $N_t = 3$ Flat-top PSF ($p = 0.223$ and $p = 0.001$, respectively), as shown in Fig. 7d, e. The fidelity of the prints was evaluated using the Jaccard index between the micro CT scans and the model. Values between 0.82 and 0.86 were obtained (Supplementary Note 15 and Supplementary Fig. 12).

We validated the possibility to fabricate different geometries, we printed additional 3D structures in acrylate resins such as a cylinder with a hole, a Bucky Ball, and a pre-Columbian object called Poporo (Fig. 8).

**Soft hydrogels.** Figure 9 shows confocal microscope images of a printed 3D structure in a cell laden hydrogel (see "methods"). It has been shown that self-healing beams such as Bessel[53] or Vortex beams[54] penetrate deeper in scattering tissue (50% longer penetration depth for skin[53]) and retain their spatial shape better than conventional Gaussian beams. An analysis of the print fidelity was performed using fluorescent images. The computed Jaccard index

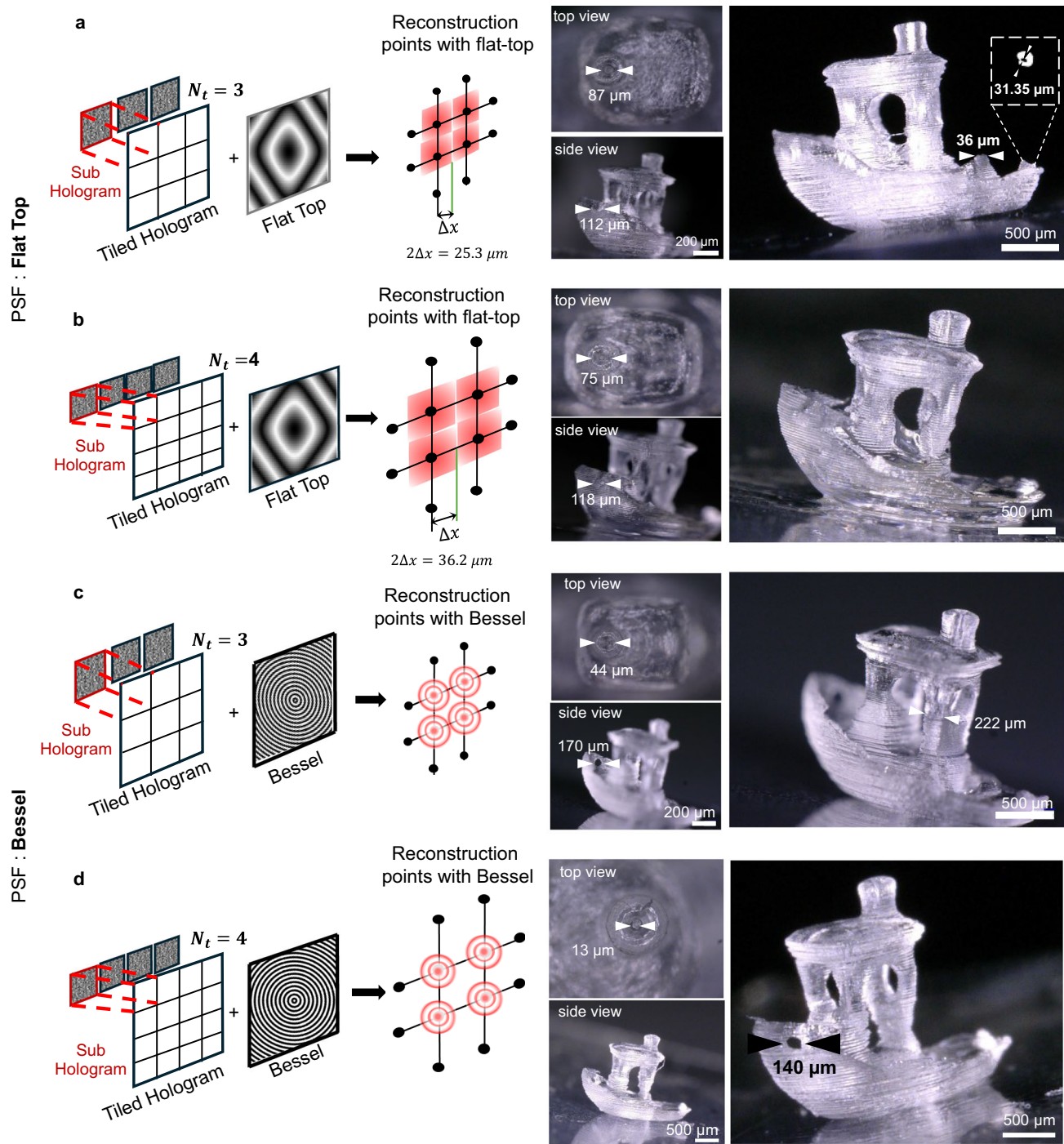

**Fig. 6 | Example of 3D printed Benchy boat fabricated with HoloVAM.** Photographs of the Benchy boat using a photoresist with TPO as photoinitiator, and TEMPO. **a** Printed part obtained using tiled holograms of 3 tiles and flat-top PSF, time multiplexed with 7 holograms per angle. The printing time is 154.92 s. Inset: Zoom in on the micro-CT scan of a 31.35 $\mu m$ hole in the back deck of the Benchy boat. Supplementary Movies 1–3 show the printing process, the micro-CT scan of the printed part, and the 3D slices from the micro-CT with size feature labels respectively. **b** Printed part obtained using tiled holograms of 4 tiles and flat-top PSF, time multiplexing with 7 holograms per angle The printing time is 146.76 sec. **c** Printed part obtained using tiled holograms of 3 tiles and Bessel PSF. The printing time is 125.976 sec **d** Printed part obtained using tiled holograms of 4 tiles and Bessel PSF. The printing time is 144.12 sec.

between 0.8 and 0.9 shows a remarkable fidelity in this scattering hydrogel (Supplementary Fig. 12). In traditional amplitude VAM with Gaussian beams such shape fidelity cannot be produced unless a digital correction of the patterns is performed[13]. Here, we have used an optical vortex PSF to exploit the self-healing properties of Laguerre-Gaussian beams in scattering media[55] (note that other self-healing beams could also be used such as Bessel

beams). The generated beam carried an orbital momentum with a topological charge equal to $\ell = 1$. The printed results show that by just exploiting the PSF properties, all details of the 3D model were printed without the pre-compensation of the projection pattern necessary for these cell-laden constructs[13]. We can also evaluate that all the features of the construct are present from Fig. 9c, and Supplementary Movie 4.

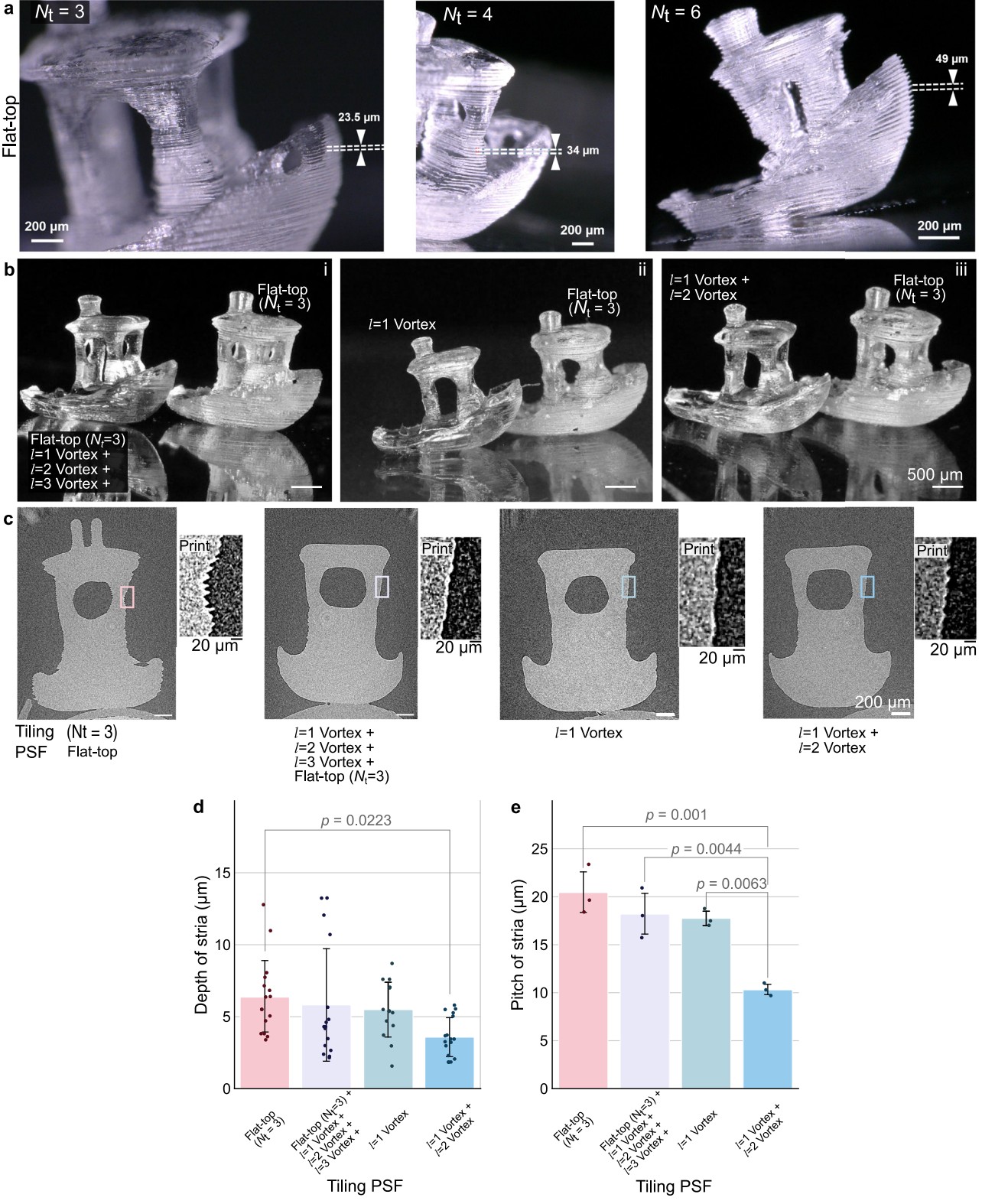

**Fig. 7 | Impact of PSF engineering and time multiplexing on sample smoothness. a** Micrographs of printed 3DBenchy boats with a flat-top as tiling PSF for different tiling numbers (3, 4, and 6). Scale bars: 200 $\mu m$. **b** Micrographs of printed 3DBenchy boats with different PSF combinations. Flat-top PSF ($N_t = 3$) is used as a benchmark which is compared to i. a. Flat-top ($N_t = 3$) + a sum of vortices with charges 1 to 3; ii. A single vortex with topological charge $\ell = 1$; iii. Two vortices, with topological charges and $\ell = 2$ and tiled with $N_t = 3$ and $N_t = 2$, respectively. Scale bars: 500 $\mu m$. The schemes of PSFs are detailed in Supplementary Fig. 8.4.). **c** Cross-sections of microCT scans of the printed parts. From these cross-sections, **d** depth and (**e**). pitch of the striations measured at $n = 3$ different sites (chimney, cabin, hull) for $n = 1$ printed boat per PSF scheme. Error bars indicate one standard deviation. Supplementary Tables 2–5 provides raw data for the graphs.

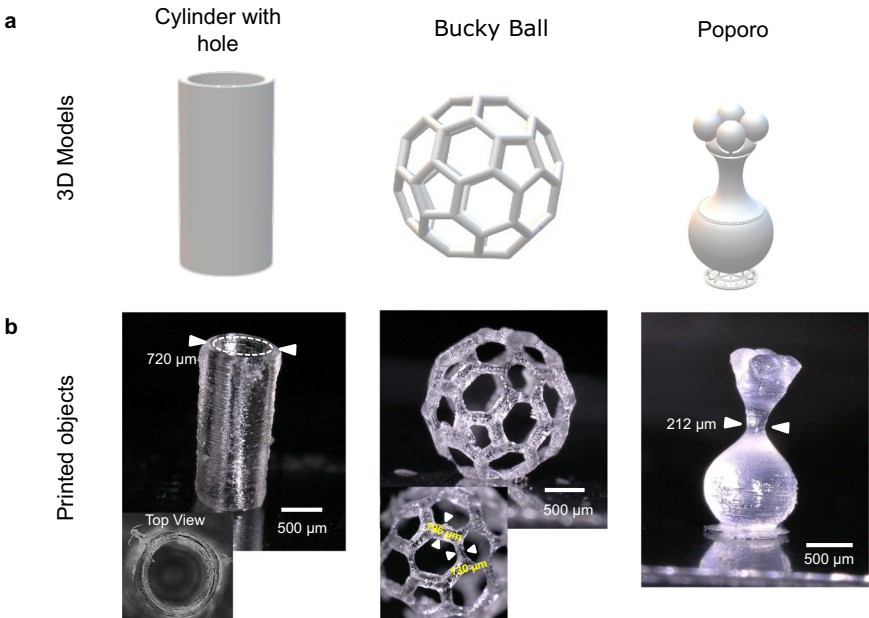

**Fig. 8 | Examples of 3D printed objects with HoloVAM. a** 3D Models. Left, cylinder with hole; Middle, Bucky Ball; Right, Poporo (a pre-Columbian object). **b** Micrographs of the obtained prints. Left, cylinder was printed in 36.96 s, using a Flat-top as a PSF and the holograms were tiled $N_t = 6$; Center, Bucky Ball printed in

37.03 s, using an optical vortex with topological charge $\ell = 1$ as a PSF; Right, Poporo printed in 29.88 s, using an optical vortex with charge $\ell = 1$ as a PSF, and the holograms were tiled $N_t = 3$. Scale bars: 500 μm.

## Discussion

In this work, we have experimentally demonstrated the use of holographic phase encoding for tomographic volumetric additive manufacturing. We show that by either rapidly projecting holograms onto the rotating photoresin or by using a single projection using "non diffracting" beams such as Bessel or vortex, we can produce low-divergence light beams. These low-divergence beams approximate the ray optics assumption of the Radon transform, the computational backbone to calculate tomographic projections. We illustrate the low divergence of these phase-encoded projections through computational simulations and experimental recordings.

Light intensity efficiency is key to fabricating objects within tens of seconds in tomographic VAM approaches, mainly because of the low concentrations of photoinitiators. Phase-encoding increases the light efficiency 28-fold compared to amplitude encoding (for the tomographic patterns to print the 3DBenchy Boat). As a result, we were able to print millimetric objects with a 40 mW cw laser source in one minute, instead of 1–3 W input optical light powers typically used in other works[9].

We also introduced a computation pipeline to compute holograms for tomographic projection. We used the HoloTile method[38,39], to provide both reduced speckles and control of the shape of individual pixels at the reconstruction plane. The addition of Fresnel lenses allowed us to synthetically produce a sequence of holograms resulting in low etendue projections. This method may open the possibility of adapting current light-based 3D printing systems based on incoherent light engines to coherent light engines that offer much higher efficiency and flexibility in light control. Compared to liquid crystal spatial light modulators, a DMD has a faster display rate, wavelength-independent and polarization-independent modulation[33,35]. A DMD can be used at shorter wavelengths, where LC-SLMs are not suitable due to degradation of liquid crystal molecules[40]. Moreover, different from conventional TVAM, the binary phase patterns are 1–bit images, which allows us to use the maximum frame rate of the DMD ~22 kHz, whereas TVAM uses 8-bit images, which limits the rate of the DMD at

290 Hz. The high frame rate of 1-bit images and the print setting of our system enables us to multiplex on time up to 36 holograms per angle.

These results are the first demonstration of tomographic phase-encoded light projection for volumetric additive manufacturing. They are an addition to the family of Vat-photopolymerization methods and may open new avenues for additive manufacturing with lower powers, more versatile light control, and simplified light engines. In Supplementary Table 1, we summarize the performance of the current volumetric additive manufacturing technologies and compare them to our method.

Phase-encoding has the potential for realizing a simpler optical setup, since a single Fourier lens is used to project the intensity reconstruction in the printing plane instead of two lenses in TVAM. Phase encoding could also correct for downstream optical aberrations, including avoiding the index matching bath used in this work. Further possibilities can be envisioned to even design a lensless projection system similarly to the systems currently in development for augmented reality holographic projection[56].

We have also printed in a scattering material using holograms without correcting the target amplitude projections thanks to PSF engineering. The use of PSF of an optical vortex generated with a helical phase plate allows printing in a hydrogel loaded with cells. The self-healing or self-reconstructing properties of the Laguerre-Gauss beams have shown the ability to penetrate deeper into scattering media[53,54]. Here, this effect helps to 3D print a high-fidelity construct.

We expect that phase coding could provide a path to increase the spatial resolution of TVAM to $\lambda/NA$, where $NA$ is the numerical aperture of the Fourier Lens. A possible strategy is to time multiplex holograms at each projection angle by adjusting the PSF hologram to produce shifted focused points within each pixel grid[48]. Given a lens of numerical aperture 0.2, the spatial resolution could be expected to reach 2 μm. However, a wave optics optimization for TVAM would need to be used to compute the projection patterns to reach such a resolution[25].

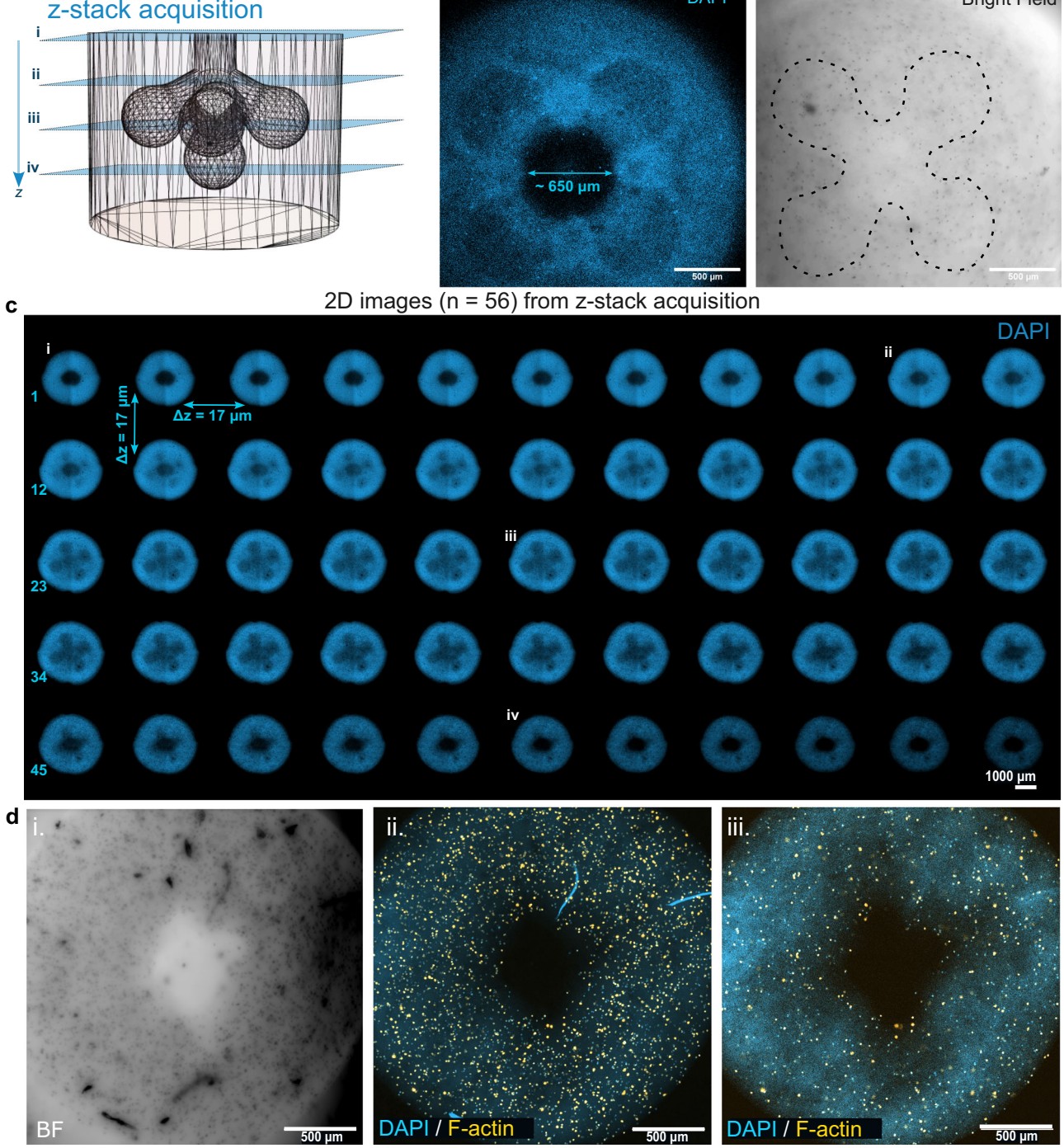

**Fig. 9 | Example of a 3D printed construct in a hydrogel loaded with cells.**
**a** Diagram from CAD design illustrating the area from which the z-stack microscopy images were acquired in the bioprinted construct. **b** Fluorescence (left) and brightfield (right) confocal images showing the multiacinar cavity inside the 3D cellularized construct. Scale bars: 500 $\mu m$. **c** Montage of 2D images from z-stack acquisition. Scale bar: 1000 $\mu m$. The Brightfield images displaying the sum projection of the 2D slices were added in the Supplementary Material. **d** Representative brightfield (i) and multi-channel fluorescence (ii, iii) microscopy images showing the maximum intensity projections of 100 slices (i, ii) and one single slice ($n = 66$). Scale bars: 500 $\mu m$.

## Methods
### Optical setup
The schematic of the experimental setup is shown in the Supplementary Fig. 1. It consists of one 40 mW CW laser diode at 405 $nm$ (OBIS 405 nm LX SF 40 mW) coupled into a single-mode fiber (Thorlabs P3-405B-FC) to obtain a good quality Gaussian beam that is collimated to best approximate the Radon transform. For

comparison, a 405 nm LED with an emission area of 1 mm² and emission angle of 170° has a space bandwidth product 10⁶ times larger than a single mode 405 nm laser diode (1 $\mu m$ × 2 $\mu m$) emission area and 30° by 15°).

The collimated beam illuminates a DMD (Vialux DLP7000: $L \times W = 768 \times 1024$ micro-mirrors and pixel pitch = 13.6 $\mu m$) placed in a Fourier configuration and aligned to meet the blazed grating criteria

for efficiency. After the Fourier lens L1 ($f_1 = 150$ $mm$), a spatial filter (SF) was added to filter out the zero-order and keep the -1 order. L2 ($f_2 = 150$ $mm$) and L3 ($f_3 = 200$ $mm$) form a 4-$f$ system conjugating the Fourier plane and rescaling the holograms on the sample plane. The DMD was synchronized with the rotary stage (Zaber, RSW60C-E03T7-KX13A) using a Data Acquisition Card from National Instruments (X series DAQ, PCIe-6321) at a typical framerate of 1600 $Hz$. The rotatory stage holding the sample vial is set to rotate at a constant speed (30°/s) while a sequence of holographic projections is displayed every $\Delta\theta = 0.6°$ synchronously on the DMD. The choice of 30 degree/s is a typical value for VAM printers. However, rotation speed could be increased as the projection rate of the DMD is 22'700 images (holograms) per second. An index-matching bath of vegetable oil is used to avoid lensing effects from the cylindrical vials containing the photoresin ($n = 1.48$).

Two inspection systems using lenses L4 ($f_4 = 75$ $mm$) and L5 ($f_5 = 150$ $mm$) with camera 1 (iDS UI307xCP-M), and lenses L6 ($f_6 = 100$ $mm$) and L7 ($f_7 = 100$ $mm$) with camera 2 (iDs UI327xCP), were implemented for the inspection of the polymerization process, and holographic projections respectively.

## Tomographic amplitude projections

First, the 3D model (typically a.stl file) is voxelated and sliced (Supplementary Fig. 6). The $y$ axis is selected parallel to the rotation axis of the vial. In tomographic printing, the Radon transform $R(r, \theta)$ is digitally computed to produce angular projections followed by the Filtered back-projections (FBP)[57].

These tomographic projections are the target intensities of the GS algorithm that generates the CGH. Projections were calculated over 360° with an angular resolution of 0.6°.

## Computer generated holograms

Computer-generated holograms (CGHs) typically suffer from speckle noise, which generates grainy images[29,35,40,58]. In the context of VAM, speckle noise needs to be avoided in the projected intensity image because they could create unwanted printed speckled voxels. To improve the reconstruction quality of the phase-only holograms, several methods such as time averaging iterative algorithms applying bandwidth constraints[44–46], and or camera in the loop[59] have been proposed. Due to the uncontrollable interference between adjacent pixels in a CGH caused by overlapping Airy disks, tiling holograms in space and/or time or by other output pixel separation methods can provide attenuation of unwanted noisy interference[30,37,38,42,47].

In general, the computational time for full-size CGHs scales as $M \log M$, where $M^2$ is the number of pixels in the image. An additional advantage of tiled holograms is the reduction of the computation time by at least an order of magnitude since the dimensions of the tiles (sub-holograms) are smaller than those of a full-size hologram and only the smaller sub-holograms need to be computed. Moreover, holographic projections give us the possibility to engineer the shape of the projected output "voxel shape" in 3D thanks to wavefront control.

Spatial Light Modulators (SLM) based on liquid crystals are typically used to modulate complex light fields; but they are not well suited for VAM due to their low frame rate (20–100 $Hz$) and reduced resistance to short wavelengths (<450 $nm$). Thus, we use a DMD, which is a binary amplitude SLM. To implement phase modulation on the DMD, we use the binary Lee hologram method[32,33]. The synthesis of diffractive phase encoding using Lee holograms has been of increasing interest in various applications thanks to the high phase modulation rate of DMDs[31,33,35,60].

## Acrylate based photoresin

**Photocurable resins.** Photoresins were prepared by mixing the photoinitiator TPO (Diphenyl (2,4,6-trimethylbenzoyl)- phosphine oxide, Sigma) in a polyacrylate commercial photoresin (PRO 21905, Sartomer) to a concentration of 3 mM using a planetary mixer deaerator

(Kurabo Mazerustar KK-250SE). We used the following protocol: 30 s at low revolution speed, and High rotatory speed, 120 s at high revolution speed, and middle rotatory speed, and 120 s for the deaeration stage (high revolution speed, and low rotatory speed). The planetary mixer has 10 predefined programs that set the number of the revolutions in 10 different speed levels from 60 g to 420 g. Each program has a different combination of revolving speed, rotation speed, and treating time (program 4 was used).

In addition, (2,2,6,6-tetramethylpiperidin-1-yl)oxidany (TEMPO) was added to the photoresist at 0.3 mM to improve the printability of small features. The photocurable resin was poured into cylindrical glass vials (outer diameter 12 mm) and then sonicated to remove air bubbles.

**Post-processing of printed parts.** Printed parts were recovered from the glass cylinders and rinsed for 10 min in propylene glycol monomethyl ether acetate (PMGEA) with slow agitation in a vortex mixer. Then, where cleaned in isopropyl alcohol (IPA) for 10 min more. Finally, they were post cured under UV light while immersed in PMGEA.

**Hydrogels.** Gelatin Metacryloyl (GelMA) was synthesized from porcine gelatin (Sigma) according to Van De Bulcke et al.[61]. Briefly, Type A bloom = 300 porcine gelatin powder (Sigma, G2500) was dissolved at 10% w/v in Phosphate Buffered Saline (PBS) 1x at 50 °C. 25 mL methacrylic anhydride (Sigma, 760-93-0) was added dropwise and stirred at 50 °C for 3 h to functionalize the gelatin. The excess unreacted methacrylic anhydride was removed by centrifugation for 15 min at 3000 rpm. Furthermore, the material was actively filtered with PBS against 12 times its volume. The solution was lyophilized for 48 h and stored at −20 °C, protected from light. 7% w/v GelMA solution was prepared by dissolving lyophilized GelMA powder in PBS containing Lithium phenyl-2,4,6-trimethylbenzoylphosphinate (LAP, Sigma-Aldrich, 900889) at 0.5 mg mL$^{-1}$, followed by filter sterilization at 40 °C. GelMA solutions were stored at 4 °C, shielded from light, for up to 2 weeks.

**Cell culture.** Human foreskin fibroblasts (HFF-1) were obtained from ATCC and cultured in Dulbecco's Modified Eagle's Medium (DMEM) without phenol red, supplemented with 1% Penicillin-Streptomycin (Gibco), 2% L-glutamine (Gibco), and 15% FBS (Gibco) under standard conditions (37 °C, 5% CO2).

**HoloVAM-bioprinting.** For HoloVAM bioprinting, a multiacinar cavity structure was fabricated. Fibroblasts were detached, counted, and centrifuged. Cells at a final density of 0.5 million cells mL$^{-1}$ were resuspended in GelMA with LAP and gently mixed using a 1000 μL pipette tip.1 mL of the GelMA + LAP + HFF-1 mixture was dispensed into ethanol-sterilized cylindrical glass vials (12 mm diameter) with hermetically sealing caps under sterile conditions in a biosafety cabinet. The vials were placed in water at 2 °C to induce gelation of the GelMA and printed within 1 h.

**Post-processing -cell-laden hydrogel.** Post-printing, the glass vials were heated to 27 °C for 5 min in a water bath. Under sterile conditions in a biosafety cabinet, pre-warmed PBS at 37 °C was gently added to the vials, and they were gently agitated to remove uncross-linked GelMA.

## Imaging

**Micro-CT scans.** Printed objects were imaged with voxel sizes of $1.7 \times 1.7 \times 1.7$ μm$^3$ under a 160 kV X-ray transmission tomography (Hamamatsu, Japan). 3D visualizations and cross sections of the pieces were obtained using Fiji-ImageJ[62].

**Confocal microscopy.** Confocal microscopy was used to image the bioprinted constructs which were fixed in formaldehyde 4% v/v in PBS

for 15 min, then rinsed twice with PBS. Samples were permeabilized with 0.2% Triton X-100 in PBS for 10 min at room temperature, washed three times for 5 min each with PBS, and incubated with ((R)-4-Hydroxy-4-methyl-Orn(FITC)[7])-Phalloidin (1:60, 0.16 nmol mL−1) in PBST + 1% BSA for 30 min at room temperature. They were rinsed with PBS three times for 5 min each, stained with DAPI in PBS (1:1000) for 5 min, washed once with PBS, and imaged using a motorized inverted confocal microscope (Leica SP8) with a 5x NA 0.15 air objective (WD = 13.7 mm, HC PL Fluotar, Leica). Images were visualized and processed using ImageJ and pseudo-colored with the colormaps freely provided by Bio Imaging & Optics Core Facility the at EPFL.

**Photography.** Printed parts were imaged with a digital microscope (VHX-5000, Keyence) with magnifications between 20 and 200x.

**Simulations.** 3D renderings of.stl files, recorded light intensities and simulated intensity distributions were produced with Wolfram Mathematica® 13.1[63]. To calculate the intensity projections using the Radon transform, the CGHs, and simulations of propagations were done using MATLAB®[64] and Wolfram Mathematica® 13.1[63].

**Experimental axial measurements of projections.** To record the 3D projections of the CGH in the build volume, camera 2 (UI327xCP, IDS Imaging) in the setup was mounted on a linear motorized micrometric stage (Z912B, Thorlabs). The servo motor was controlled and synchronized with the camera acquisition so that the camera would record a photograph every 20 μm.

**3D models.** The 3D.stl models of the #3DBenchy Boat (3DBenchy.com; licensed under CC BY-ND 4.0) and custom-made gear, cylinder, and pancreatic acinus were used. We also used the 3D model of the C60 Fullerene Buckyball by Almateus on cults3d.com (licensed under CC BY-ND 4.0), the 3D model of the Poporo Quimbaya v3 on Thingiverse.com by 3dtensorial (licensed under Creative Commons – Attribution).

**Hardware control.** To drive and synchronize the devices in the implemented printed, we used Python as a programming language. To control the DMD we used the ALP4lib module of Sébastien M. Popoff[61]. For the rotatory motor stage, we used the Zaber library. (https://www.zaber.com/software/docs/motionlibrary/ascii/references/python/). We used a National Instruments DAQ card (PCIe-6321, X Series) as a reference clock to trigger the sequence display on the DMD and the cameras 1 and 2.

**Statistical analysis.** One-way ANOVA tests were performed with Microsoft Excel's Data tool with alpha values of 5%. Ad-hoc Tukey tests were performed using SciPy's (https://docs.scipy.org/doc/scipy/reference/generated/scipy.stats.tukey_hsd.html).

## Data availability
The data supporting the results of this study are available in the Supplementary Information and Supplementary Data file.

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

## Acknowledgements

C.M. received funding from the Swiss National Science Foundation under project number 196971 - "Light based Volumetric printing in scattering photoresins". J.G. is supported by the Novo Nordisk Foundation, Denmark (Grand Challenge Program; NNF16OC0021948). C.M. and J.G. received funding from the Eurostars-3 (VOLTA-E!3908) joint program with co-funding from the European Union's Horizon Europe research and innovation program, Innosuisse (Swiss Innovation Agency), and Innovation Fund Denmark (IFD). The authors would like to acknowledge Gary Perrenoud, Lionel Pittet, and Albert Taureg (PIXE Platform, EPFL) for their support with microCT imaging of the printed structures.

## Author contributions

M.I.A.C. Conceptualization. Performed experiments, data acquisition, data interpretation, validations, and simulations. Writing: Original draft, review and editing. A.G.M. performed experiments, data acquisition, and simulations. J.M.W. Data Curation, data interpretation, validation, and visualization. Writing: Original draft, review, and editing. V.S. performed the cell laden GelMa sample preparation. Performed experiments, optical imaging, and data curation. A.B. Methodology. Writing: review and editing. J.G. Conceptualization. Methodology. Supervision. Writing: review and editing. C.M. Conceptualization. Methodology. Supervision. Writing: Original draft, review and editing.

## Competing interests

J.M.W. is currently an employee of Readily3D (Switzerland), a company that develops and commercializes tomographic volumetric 3D printers. C.M. is a shareholder in Readily3D. J.G. is inventor on a PCT patent application related to this work titled "Holographic Volumetric Additive Manufacturing" (PCT EP2024/080265). All other co-authors declare no other competing interests.
