## [Transparent Peer Review file · Nature Communications]

Holographic Tomographic Volumetric Additive Manufacturing

Corresponding Author: Ms Maria Alvarez-Castaño

Version 0:

Reviewer comments:

Reviewer #1

(Remarks to the Author)

The authors report a novel tomographic volumetric additive manufacturing technology combined with holographic phase modulation. The method can achieve a resolution of 50 μm in negative feature printing by utilizing a low-power laser light source. The findings indicate that the proposed volumetric 3D printing technique holds broad applications, enabling the rapid fabrication of microstructures. This paper is well-written; however, the impact, originality and contribution need to be better demonstrated to match the quality of nature communication journal. A few significant concerns:

1. In this volumetric 3D printing, the printing size and resolution are limited. What's the improvement of the proposed technology in size and resolution compared to other volumetric printing techniques? A table is suggested to summarize current volumetric printing methods and compare them to your proposed printing technology, including printing size, resolution, accuracy, speed, light source, and materials.
2. The benefit of holographic projection (phase modulation) is not experimentally justified compared to conventional image projection (amplitude modulation). Instead, holographic projection has severe drawbacks: (1) the speckles generate rough surfaces with obvious steps. Such steps are much worse and more difficult to remove than the pixels in raw DMD patterns. (2) Besides, what's the error of the generated hologram compared to the required pattern? The GS algorithm will never give an error-free pattern. What's the impact of the hologram error on the printing results? Instead, the image projection can be accurately controlled with the exact intensity pattern.
3. In the result section, only a single millimetric sample was printed. More samples are needed to demonstrate the printing ability. Some simple and small features, like pillars and channels, are needed to quantify the printing resolution. Complex structures, such as lattice structures, are also suggested.
4. It is not a fair comparison for the power consumption or the printing time, since the printed size (mm-scale) is much smaller than literature (cm-scale). If scaling the printing size by 10x, then the needed power for an image with the same light intensity will be 100 times larger than now, which is still around a few watts, comparable to existing ones.
5. Have the authors compare the dimensions of the printed sample with the designed sample. It will be more convincing if demonstrate this. Any compensations for model design in sample printing?
6. Compared to the recently published upgraded TVAM, the printed sample has many obvious staircases. Can you explain why in detail? How can this effect be reduced or avoided? Can the proposed method print smooth surfaces for more applications?
7. There are many typing errors in supplementary material. Please double-check the grammar and the format. Such as: was display; as a the target; a intensity reconstruction; fig. 2b; Fig 3.1; Here we is possible.
8. In this paper, the photocurable resin was used by mixing the photoinitiator and a commercial resin. What is the mixing rate? Does this rate affect printing results? Any other materials were printed to verify the effectiveness of this method. A 40mW laser is used for printing. Would it be suitable for other resins for broad applications?
9. Holographic Volumetric 3D printing can be found in some published papers, which utilize the interference of three

holographic beams to create a 3D hologram. This proposed technique belongs to tomographic volumetric additive manufacturing. Holographic Tomographic Volumetric Additive Manufacturing as the title or term may be more accurate.

Reviewer #2

(Remarks to the Author)

Review of "Holographic Volumetric Additive Manufacturing" by Álvarez-Castaño et al.

High Level Overview

This work introduces a new volumetric 3D printing technique in which holographic light patterns are used to fabricate parts in a liquid photoresin. In contrast to tomographic volumetric additive manufacturing (TVAM) that produces light patterns through amplitude modulation, the method demonstrate here uses phase modulation and the Lee hologram method for light pattern creation. As such, the authors call this method holographic volumetric additive manufacturing (HoloVAM) and show remarkable print quality of a 3D Benchy test print. In particular, they demonstrate millimetric scale printing with micron-scale feature sizes. The increase in print resolution can be attributed to the use of self-healing Bessel beams in conjunction with time-multiplexing to transmit patterns at different planes within the print volume. Furthermore, it is shown that HoloVAM is 10-fold more light efficient than TVAM since all pixels on the spatial light modulator can contribute to light pattern creation.

Validity

Interpretation of the data was well done, and conclusions justified and reasonable. My main suggestion is to incorporate the microCT data into the analysis to further improve the quality of the manuscript and support their claims.

Significance

The addition of holographic methods to tomographic volumetric printing will open new frontiers in this printing modality. As such, I expect this paper to be of high impact and play a transformative role in the additive manufacturing landscape.

High Level Comments

1. The paper provides introduces a new volumetric 3D printing approach which improves upon TVAM via increasing the printable volume (Bessel beams + multiplane projection) and increasing light efficiency. The authors also state that phase-encoding uses a simpler optical setup (than TVAM). While I agree that in its simplest form HoloVAM does use a simpler optical configuration, the authors motivate the strength of their approach using a more complicated system. I.e. using Fresnel lenses to achieve multiplane projection and an Axicon to achieve Bessel beams, both of which adds complexity beyond that of TVAM. Prior literature has shown TVAM capable of making parts with comparable resolution to HoloVAM without the need for either Fresnel or Axicon components. I recommend that the authors either remove or rephrase this statement.
2. An index-matching bath surrounds the print volume in HoloVAM. In TVAM, it has been demonstrated that this bath can be removed via a suitable remapping of the Radon coordinates or explicitly using optical ray tracing. Given this information, could HoloVAM be expanded to not use an index-matching bath to further simplify the system? Further, could this method be expanded to a non-telecentric (e.g. cone beam) configuration?

Detailed Comments

1. On Line 95, "...reach negative features (holes) of 50 μm reliably". I do not agree with the "reliable" part of this statement. A series of 3D Benchy parts are shown in Figure 6 for different tile number and PSF. In particular, feature sizes for the chimney, cargo box, and anchor hole are shown, and vary widely depending on PSF and tile number. Of the four parts shown, only Fig. 6 c,d) show negative features less than 50 μm and furthermore, are much smaller than the target size of 97 μm . Has a repeatability study been performed to assess print-to-print variation?
2. Line 406: MicroCT imaging. Based on your MicroCT imaging, can you quantify the printed part accuracy beyond that of the microscope images? E.g. 3D Jaccard similarity, signed-distance field, etc.
3. Line 340. Is the rotation rate of the vial (30 deg/s) chosen based on a limitation, such as the framerate needed for multiplane projection or non-laminar fluid flow in the vial?
4. Discussion. Another benefit of binary-pattern projection is increased projection frame rate. Binary images take up less memory than typical gray-scale images in TVAM (e.g. 8-bit).
5. How similar is the refractive index (RI) of the vegetable oil to your photocurable resin ($n \sim 1.47$)? Based on the stated number of significant digits I assume your RI is between [1.465, 1.474] depending on method of rounding. If the RI wasn't perfectly matched, I would expect a magnification (or demagnification) of your projected field. Do you observe any such effect? At such small feature scales (10s of μm), would a RI error of ± 0.005 play a significant role on your print fidelity?

Sentence-level clarity improvements

1. Line 15. Replace "into a resin" to "into a photo-curable resin" or "into a photoresin".
2. Line 183. "There is a clear trend of decreasing the error of the projected hologram while the PSNR increases standing a large speckle noise reduction by performing time multiplexing." Please consider rephrasing as it is currently confusing to read. I understand this statement as: "For increasing number of tiles, MSE and speckle noise decreases while PSNR increases, with further improvement with time-multiplexing."
3. Line 332. "L X L = 768 X 1024 ...". Do you mean "L X W = 768 X 1024" ?

Formatting improvements

Body

1. General comment about terminology consistency. The terms "resin" and "photoresin" are used interchangeable. Please be

consistent with your usage and choose a single term.

2. Line 41. In this paragraph tomographic volumetric additive manufacturing is given the acronym TVAM, but yet later in the paragraph is still explicitly written out.
3. Line 73. "We term this novel technique HOLOVAM". Font size appears larger than body.
4. Line 147. Add space between "Supplementary Fig".
5. Line 223. Replace "With the aim of use a light engine base on" with "With the aim of using a light engine based on".
6. Line 228. Replace "We exploit the PSF engineering reducing" with "We exploit the PSF engineering by reducing"
7. Line 231. Replace "non-diffractive beams, Bessel beam produce a" with "non-diffractive source, such as a Bessel beam, produces a"
8. Line 239: Add space between "10mm".
9. Line 267: Add space between "Ntwere"

Figures

4. Figure 2b: Histogram of pixel gray level has x-axis range from [0,1]. However, on Line 123 gray level is defined as ranging from [0,255]. Either change Line 123 to match Figure 2, or vice-versa.
5. Figure 2c: Does the phase pattern include the PSF?
6. Figure 4c, caption: Replace "Bessel PDF" with "Bessel PSF"

References

1. Please make sure references adhere to Nature formatting guidelines (<https://www.nature.com/nature/for-authors/formatting-guide>). Several references are listed with more than 5 authors (e.g. References 8,9) which should be replaced with [First Author] et al.

Reviewer #3

(Remarks to the Author)

The manuscript presents what the authors call holographic volumetric additive manufacturing in which they explore phase rather than amplitude approach to Tomographic Volumetric Additive Manufacturing (TVAM). They use a binary amplitude spatial light modulator in the form of a micromirror array (DMD) to create a phase hologram with the Lee method that is then optically reconstructed in the resin. As written, the manuscript is at best confusing, which leaves me recommending rejection or possibly major revision because it is difficult to evaluate the authors' approach and results. Some comments and examples are included below.

Questions/comments:

- Fig. 1 claims to show the optical set up of the system. But later in the text Fig. S1 is referred to as the actual optical set up. Unfortunately, the two are quite different and cause significant questions. For example, Fig. 1 does not show diffraction order spatial filtering, which is crucial in Lee hologram method to convert binary amplitude modulation from the DMD to phase modulation. However, diffraction order spatial filtering is present in Fig. S1. Why the conflicting figures of the system?
- Relatedly, the Fig. S1 caption says, "A single lens allows the reconstruction of the projected holograms into the rotatory resin container...", but Fig. S1 shows that there are actually several lenses.
- The authors state that the laser diode is single mode, which I assume refers to its spectral properties. What is the actual spectral width of the source and therefore its coherence length? How does the coherence length compare to the critical lengths in the hologram reconstruction and is the necessary degree of coherence maintained?
- Eq. 1 indicates that the authors use grayscale images on the DMD. Such images are generated through temporal modulation of the fraction of the time a given pixel is "on" during a frame time. What are the effects in the image region in the resin container when at any given instant in time only some of the pixels are on, and in the next instant some of them have turned off while other pixels have turned on in order to hit the graylevel for each of the pixels, which is determined automatically in the DMD drive chip? The authors analysis implicitly asserts that this is the same situation as having the light amplitude from each pixel be continuously on at the proper graylevel for the entire frame time. This seems problematic, but at a minimum should be justified and confirmed.
- What's the minimum exposure any given small region of resin receives? If it is zero, how is this explained, and how does this compare to amplitude TVAM?
- For the Lee method implementation, what is the trade-off the authors have made between resolution and fidelity of the generated phase pattern based on choice of carrier spatial frequency, spatial sampling in terms of how many pixels per macro-pixel, and the size of the iris used for spatial frequency filtering for the -1 order? Given the limited number of pixels on the DMD, what does this imply about the size and resolution of what can be fabricated in the resin?

Reviewer #4

(Remarks to the Author)

Version 1:

Reviewer comments:

Reviewer #1

(Remarks to the Author)

Thanks for the revision. However, I still don't understand why holographic projection is used to replace simple image projection. Lee's method was used to smoothen the holographic projection, but still, it is not comparable to the image projection quality. This can be obviously noticed from the printed samples, whose surface is very rough and even the shapes are significantly deformed. I was not convinced by the novelty and quality of this paper as NC journal.

Reviewer #2

(Remarks to the Author)

We appreciate the author's thorough response to all questions raised. As such, we recommend the manuscript for publication.

Reviewer #3

(Remarks to the Author)

The authors have adequately addressed the concerns raised in my original review.

Reviewer #4

(Remarks to the Author)

Response to reviewers

Holographic tomographic volumetric additive manufacturing

We thank the editor for the handling of our manuscript. We hereby address all the reviewers' comments and criticisms.

We hope that the resulting manuscript will now be ready for publication in Nature Communications.

In our responses, we highlighted the changes made in the manuscript with blue color.

REVIEWER COMMENTS

Reviewer #1 (Remarks to the Author):

The authors report a novel tomographic volumetric additive manufacturing technology combined with holographic phase modulation. The method can achieve a resolution of 50 um in negative feature printing by utilizing a low-power laser light source. The findings indicate that the proposed volumetric 3D printing technique holds broad applications, enabling the rapid fabrication of microstructures. This paper is well-written; however, the impact, originality and contribution need to be better demonstrated to match the quality of nature communication journal. A few significant concerns:

Reply: We thank the reviewer for the comments and thoughtful questions. We answer the questions below in green and the modifications on the manuscript are in blue.

1. In this volumetric 3D printing, the printing size and resolution are limited. What's the improvement of the proposed technology in size and resolution compared to other volumetric printing techniques? A table is suggested to summarize current volumetric printing methods and compare them to your proposed printing technology, including printing size, resolution, accuracy, speed, light source, and materials.

Reply: Thank you to the reviewers for their suggestion. We added the following comparison table in Supplementary Information S10 (Supplementary Table 1) that summarizes the main parameters of current volumetric printing techniques.

[REDACTED]

To clarify the improvement of our proposed method in terms of print size and resolution, it is important to highlight the efficiency of the light engine and resolution. Our work uses a low power single mode diode laser coupled to a single-mode fiber that provides 13.2 mW of optical power. To illustrate our method, we produce $2 \times 2 \times 2 \text{ mm}^3$ 3D structures with $25 \mu\text{m}$ feature size. This printing volume and feature size would not have been possible to produce with amplitude modulation using the latter optical power (due to the low effective power efficiency of the amplitude projection light engine).

Print volume: the print volume is directly related to the optical power used and can thus be extrapolated to larger print volumes. For example, the TopWave 405 nm is a single mode laser from TOPTICA Photonics with 1 W optical power. This would result in a printed volume of $12 \times 12 \times 12 \text{ mm}^3$ with the same minimum feature size ($25 \mu\text{m}$) (assuming 500 mW in a single mode -SM- fiber) across the build volume with the same build time. By using phase-only SLMs instead of DMDs (the latter converts binary amplitude modulation into an analog phase modulation), an additional factor up to ~ 9 times in light engine efficiency could be gained and therefore yield structures of volume $36 \times 36 \times 36 \text{ mm}^3$ could be printed with minimum feature size $25 \mu\text{m}$. Due to the wavelength used in this study (405 nm), we used DMDs as they are more robust to this light wavelength than phase liquid crystal SLMs, which tend to age when operated at this wavelength. Other wavelengths such as green where photoinitiators exist could make use of phase SLMs. The method described in our manuscript can be implemented with any SLMs.

In addition, as is well-known in TVAM, it is possible to further increase the volume by sacrificing build time (i.e longer build time) while including the effect of diffusion and sedimentation.

2. The benefit of holographic projection (phase modulation) is not experimentally justified compared to conventional image projection (amplitude modulation). Instead, holographic projection has severe drawbacks: (1) the speckles generate rough surfaces with obvious steps. Such steps are much worse and more difficult to remove than the pixels in raw DMD patterns. (2) Besides, what's the error of the generated hologram compared to the required pattern? The GS algorithm will never give an error-free pattern.

Reply: In amplitude modulation, the amplitude projection is well defined at the focus, which for VAM is the center of the vial or sample holder. However, away from the plane of focus, the projection becomes blurred due to beam divergence. This results in a loss of resolution/fidelity of the resulting 3D print away from the plane of focus. In contrast, phase modulation allows axial control of the projections. This axial control can produce highly collimated tomographic projections, resulting in a 3D object with the same resolution throughout the volume. Another advantage of phase modulation over amplitude modulation is the efficiency of the light engine. For both amplitude modulation and phase modulation (this work), a binary reflective SLM based on micromirrors (DMD) is used. We have measured that the light efficiency of phase coding is up to 20 times more efficient than amplitude coding (as explained in Supplementary Material S3). For phase-only SLMs, like liquid crystals, the light efficiency further increase.

As the reviewer pointed out, speckles can result from using phase modulation. It is this drawback that is specifically addressed in this manuscript. We use tiled holograms and point spread function (PSF) engineering, known as HoloTile, to provide speckle noise reduction. The "steps" or "striations" are not related to the speckles but rather to the highly collimated projected patterns that give rise to waveguided light and/or to self-focusing. Note that the latter is also seen in amplitude coded VAM. As we have discussed in the manuscript, the periodicity of the striations is directly related to the pixelated reconstructed tiles. We have shown experimentally that using three different PSF engineering, we can reduce the striations amplitude and even eliminate it when a hydrogel resin is used (see answer to question 8).

Concerning the pattern projection fidelity, we have quantified the fidelity by using two metrics: 1. Mean squared error (MSE) and 2. Structural similarity index (SSMI) between the ground truth and experimental projection.

We have added the MSE and SSMI graphs in the Supplementary Information S5.1

and the following text:

“We use two different metrics to quantify the CGH projection error: the Mean squared error (MSE) and Structural similarity index (SSIM) offer complementary insights into the projection fidelity when speckle noise is present. To cover a projection depth of 4mm, we evaluate the error as a function of an increasing number of projecting planes from 4 to 34. With 34 projections, a CGH is optimized every 117.6 mm whereas for 4 projections, a CGH is optimized every 1 mm. We observe that the uniformity of the error across the propagation depth increases with the number of CGHs but that the fidelity as measured by SSIM and MSE decreases with increasing CGHs. A tradeoff exists between error uniformity across the build volume and fidelity. We have selected 6-8 CGHs (corresponding to a sampling of z 500 mm – 666mm) which is a good compromise between fidelity and error uniformity.”

Supplementary Fig. 7.1. **a** Mean square error (MSE) and Structural similarity index (SSIM) of the simulated intensity reconstruction of the tiled holograms as a function of depth z in the vial compared with the target amplitude tomographic patterns for a number of tiles $N_t = 3$ (left) and $N_t = 4$ (right) using a Flat-top PSF. **b** MSE and SSIM measured from simulated intensity reconstructions of tiled holograms using Bessel PSF, compared with the amplitude tomographic pattern as a target. The simulation was made for the projected angle $\theta = 0^\circ$.

- What's the impact of the hologram error on the printing results? Instead, the image projection can be accurately controlled with the exact intensity pattern.

Reply: The error between the projected hologram and the targeted pattern has been addressed and quantified (see answer to the question above). The impact of this error on the printing results has also been quantified. For example, the height and perimeter of multiple prints of a cylinder has been measured. The fidelity depends on the optical dose and printing time. This is like traditional amplitude VAM. We show that it is possible to achieve near 100% fidelity when the optical dose is optimized (see new figure 8 below and Reviewer 2 - Detailed comments question #1). Hologram error manifests also in the stria. We analyzed several PSF and time multiplexing that minimizes stria (see answer to question 6). We also quantified the fidelity of a 3D printed complex shape (a pancreatic acinar structure, see answer to question 8 and new figure 9) using GelMa, a hydrogel, which is a different material than acrylate. Here, striation effects are much less severe since the refractive index difference between the photo crosslinked GelMa and not photo crosslinked GelMa is one order of magnitude less than for acrylate. The fidelity of the 3D printed shape (measured by confocal microscopy) and the 3D target was quantified. Moreover, we quantify the fidelity performing a similarity Jaccard index measurement, Reviewer 2 - Detailed comments question #2.

3. In the result section, only a single millimetric sample was printed. More samples are needed to demonstrate the printing ability. Some simple and small features, like pillars and channels, are needed to quantify the printing resolution. Complex structures, such as lattice structures, are also suggested.

Reply: We printed additional structures in acrylate resins such as a cylinder with a hole, a Buckyball, and a pre-Columbian object called a Poporo shown below.

We added the new figure below and the following text to the main manuscript:

“We validated the possibility to fabricate different geometries, we printed additional 3D structures in acrylate resins such as a cylinder with a hole, a Bucky ball, and a pre-Columbian object called a Poporo (Fig. 8)”.

Fig 8. Examples of 3D printed objects with HoloVAM. **a)** 3D Models. Left, cylinder with hole; Middle, BuckyBall; Right, Poporo (a pre-Colombian object). **b)** Micrographs of the obtained prints. Left, cylinder was printed in 36.96 seconds, using a Flat-top as a PSF and the holograms were tiled $N_t = 6$; Centre, Bucky Ball printed in 37.03 seconds, using an optical vortex with topological charge $\ell = 1$ as a PSF; Right, Poporo printed in 29.88 seconds, using an optical vortex with charge $\ell = 1$ as a PSF, and the holograms were tiled $N_t = 3$. Scale bars: 500 μm .

We also printed another model in a cell-laden hydrogel, see answer #7.

4. It is not a fair comparison for the power consumption or the printing time, since the printed size (mm-scale) is much smaller than literature (cm-scale). If scaling the printing size by 10x, then the needed power for an image with the same light intensity will be 100 times larger than now, which is still around a few watts, comparable to existing ones.

Reply: In amplitude projections for conventional TVAM, only a few pixels contribute to the dose required to build a volume due to the sparsity of the projection patterns. In contrast, thanks to the nature of the light engine used for phase projection (interference), all the pixels in the display contribute to the dose delivered to the resin. The increased in light efficiency was measured experimentally to be 20% (compared with amplitude projectors) which allowed to print 2x2x2 mm size object with 13.2 mW optical power. If the printing size is scaled by 10X, the amount of power would be 1.3 W which is less than current DMD system ($\approx 6\text{W}$). True phase SLMs are more efficient (up to 10 times) than the binary amplitude DMD used in this study which would then further decrease the amount of power. With a phase SLM, we computed (see answer to question above) that 500 mW optical power would be sufficient to

print an object of size 32x32x32 mm which is much lower than the few Watts used in previous studies.

5. Have the authors compare the dimensions of the printed sample with the designed sample. It will be more convincing if demonstrate this. Any compensations for model design in sample printing?

Reply: We did not include compensation for model design. To quantify printed dimensional errors, the height and perimeter of multiple prints of a cylinder with a hole in its middle has been measured (See Detailed Comments question #1 from Reviewer 2.). The fidelity depends on the optical dose and printing time. This is like traditional amplitude VAM. We show that it is possible to achieve near 100% fidelity when the optical dose is optimized.

We have printed a complex acinar pancreatic structure in a hydrogel containing cells and measured quantitatively its 3D shape by confocal microscopy which we compared with the ground truth design. (see response to question 8)

6. Compared to the recently published upgraded TVAM, the printed sample has many obvious staircases. Can you explain why in detail? How can this effect be reduced or avoided? Can the proposed method print smooth surfaces for more applications?

Reply: The "staircases" or "striations" are a well-known effect in conventional TVAM and have been independently studied and compensated for in various publications. As mentioned in our reply to question 2), the "staircases" are caused by the highly collimated projections achieved in HoloVAM thanks to the axial beam control. As discussed in the manuscript, in our case the "striations" has a periodicity that is directly related to the periodicity generated by the tiles. The inter stria distance is much smaller than in conventional TVAM (tens of micrometers versus hundreds of micrometers).

Techniques such as Blurred TVAM¹ and UV flooding of the latent image² can also be applied to the approach in our manuscript to correct the "staircases". It is important to notice that those techniques reduce the striations at the cost of resolution and fidelity. We showed a reduced striation effect by exploring PSF engineering and time multiplexing. We used an optical vortex as an additional PSF shaping technique due to its characteristic intensity distribution in a doughnut-like shape. To reduce the striation, we applied three different methods: first, we print using projections where the holograms were tiled with $N_t = 3$ convolved with a single optical vortex PSF (see Supplementary Fig. 8). Second, we multiplex in time two different PSFs (vortex with charge $\ell = 1$ and $\ell = 2$) and two different tiles, $N_t = 3$ and $N_t = 2$, respectively. Thirdly, we multiplexed in time 4 different PSF, one flat-top and three vortices with topological

charge from $\ell = 1$ to $\ell = 3$. The hologram was tiled with $N_t = 3$. Supplementary Fig. 8.4 illustrates a diagram of the three different processes.

The following new Supplementary Fig. 8.4 has been added to the supplementary:

Supplementary Fig. 8.4. PSF multiplexing to reduce striation. (Left) projections using sub-holograms tiled $N_t = 3$, convolved with a helical phase plate with charge $\ell = 1$. (Middle) time multiplexing different PSFs: top, $N_t = 3$ with a helical phase plate with charge $\ell = 1$ and bottom $N_t = 2$ with a helical phase plate with charge $\ell = 2$. (Right) time multiplexing four different PSFs.

Our new results are shown in Fig. 7 b. For comparison, we use the Benchy boat from Fig. 6 a) as a benchmark to show the stria reduction. In addition, the striations are less pronounced when TEMPO is not added to the photoresin, but at the expense of resolution and fidelity. A statistical analysis was performed to determine the stria reduction. The microCT scans were used to measure stria depth and stria pitch. For statistical purposes, measurements were taken on the chimney, hull and cabinet. The results show that the multiplexing of different tiles and vortices with different topological charge result in stria reduction.

We added the new results of Fig. 7 and the following text in the main manuscript:

“Stria reduction was quantified experimentally by modifying the point spread function of the CGH and adding time multiplexing (see Supplementary Fig. 8.4). We show that it can drastically reduce stria (Fig. 7b). The parts printed with PSFs other than the $N_t = 3$ Flat-top

exhibit a smoother, shinier surface and are more transparent. We used microCT scans of the printed parts to obtain high-resolution images of their surface as shown in Fig. 7.c. The insets illustrate the side of the cabin of the boats, a region where striation occurs. The single $N_t = 3$ Flat-top PSF resulted in a rougher print. We measured the depth and the pitch of striation at the hull, the cabin, and the chimney from these microCT scans, and observed that tiling two vortices yielded prints with significantly shallower and smaller striation than using a single $N_t = 3$ Flat-top PSF ($p = 0.223$ and $p = 0.001$, respectively), as shown in Fig. 7. d-e. Additionally, tiling with different PSF combinations also impacts print fidelity, as seen in Supplementary Fig. 12.”

Fig 7. Impact of PSF engineering and time multiplexing on sample smoothness. **a.** Micrographs of printed 3DBenchy boats with a flat-top as tiling PSF for different tiling numbers (3, 4, and 6). **b.** Micrographs of printed 3DBenchy boats with different PSF combinations. Flat-top PSF ($N_t = 3$) is used as a benchmark which is compared to i. a. Flat-top ($N_t = 3$) + a sum of vortices with charges 1 to 3; ii. A single vortex with topological charge $\ell = 1$; iii. Two vortices, with topological charges and $\ell = 2$ and tiled with $N_t = 3$ and $N_t = 2$, respectively. The schemes of PSFs are detailed in Supplementary Fig. 8.4. **c.** Cross-sections of microCT scans of the printed parts. From these cross-sections, **d.** depth and **e.** pitch of the striations measured at $n = 3$ different sites (chimney, cabin, hull) for $n = 1$ printed boat per PSF scheme. Error bars indicate one standard deviation.

Supplementary Tables 2 - 4 show the statistical analysis used to generate Figures 7.d-e. These are included in Supplementary Information S12.

S12. Stria reduction with PSF multiplexing

Statistical analysis. All values with physical units are shown in microns.

Supplementary Table 2. Results of ANOVA for differences across groups for stria Depth

Anova: Single Factor

Stria Depth

SUMMARY

Groups	Count	Sum	Average	Variance
L=1 vortex	14	76.85	5.49	3.90
L=1 vortex + L=2 vortex	17	60.81	3.58	1.94
Flat-top	16	101.74	6.36	6.88
L=1 vortex + L=2 vortex + L=vortex + Flat-top	16	93.04	5.81	16.28

ANOVA

Source of Variation	SS	df	MS	F	P-value	F crit
Between Groups	73	3	24.362	3.349	0.025	2.761
Within Groups	429	59	7.275			
Total	502	62				

Supplementary Table 3. Results of ANOVA for differences across groups for stria Pitch

Anova: Single Factor

Stria pitch

SUMMARY

Groups	Count	Sum	Average	Variance
L=1 vortex	3	53.23	17.74	0.82
L=1 vortex + L=2 vortex	3	30.93	10.31	0.43
Flat-top	3	61.40	20.47	6.71
L=1 vortex + L=2 vortex + L=vortex + Flat-top	3	54.65	18.22	6.77

ANOVA

Source of Variation	SS	df	MS	F	P-value	F crit
Between Groups	175	3	58.4	15.9	0.00099	4.1
Within Groups	29	8	3.7			
Total	205	11				

Supplementary Table 4. Results of Tukey HSD ad-hoc tests for differences across groups for stria depth

pairwise_tukey HSD

Pitch

Multiple Comparison of Means - Tukey HSD, FWER=0.05

=====

group1 group2 meandiff p-adj lower upper reject

```

-----
FT L=1V + L=2V  -10.1568  0.001 -15.1743 -5.1392  True
FT L=1-3V+FT   -2.2519  0.5124 -7.2695  2.7656  False
FT  L=1V        -2.724  0.3665 -7.7415  2.2936  False
L=1V + L=2V L=1-3V+FT  7.9048  0.0044  2.8873 12.9224  True
L=1V + L=2V  L=1V   7.4328  0.0063  2.4153 12.4503  True
L=1-3V+FT  L=1V   -0.472  0.9 -5.4896  4.5455  False
-----

```

Depth

```

=====
group1  group2  meandiff  p-adj  lower  upper  reject
-----
FT  L=1V + L=2V  -2.7813  0.0223  -5.2651  -0.2974  True
FT  L=1-3V+FT   -0.5438  0.9  -3.065  1.9774  False
FT  L=1V        -0.8693  0.7918  -3.479  1.7405  False
L=1V + L=2V L=1-3V+FT  2.2374  0.0918  -0.2464  4.7213  False
L=1V + L=2V  L=1V  1.912  0.2133  -0.6616  4.4857  False
L=1-3V+FT  L=1V  -0.3254  0.9 -2.9351  2.2843  False
-----

```

As we mentioned in the answer to question #8. We have printed in a cell-laden hydrogel as an alternative application of HoloVam where the striations were not visible (due to the material).

7. There are many typing errors in supplementary material. Please double-check the grammar and the format. Such as: was display; as a target, a intensity reconstruction, Fig. 2b; Fig. 3.1; Here we is possible.

Reply: We have corrected all typing errors, grammar and format.

8. In this paper, the photocurable resin was used by mixing the photoinitiator and a commercial resin. What is the mixing rate? Does this rate affect printing results? Any other materials were printed to verify the effectiveness of this method. A 40mW laser is used for printing. Would it be suitable for other resins for broad applications?

Replay: To obtain a homogeneous solution, we followed a protocol using the planetary mixer/deaerator KK-250SE to mix the photoresin. This device allows the mixing and deaeration of highly viscous liquids without the formation of bubbles, using a combination of revolving motion and rotatory motion. The rotation speed is adjusted according to the characteristics and type of material used. The program used consists of three different stages: 30 seconds of the Pre-mixing stage (low revolution speed, and High rotatory speed), 120 seconds of the mixing stage (high revolution speed, and middle rotatory speed), and 120 of the deaeration stage (high revolution speed, and low rotatory speed).

The following sentence was added to the manuscript:

“Photoresins were prepared by mixing the photoinitiator TPO (Diphenyl (2,4,6-trimethylbenzoyl)- phosphine oxide, Sigma) in a polyacrylate commercial photoresin (PRO 21905, Sartomer) to a concentration of 3 mM using a planetary mixer deaerator (Kurabo Mazerustar KK-250SE). We used the following protocol: 30 seconds at low revolution speed, and High rotatory speed, 120 seconds at high revolution speed, and middle rotatory speed, and 120 seconds for the deaeration stage (high revolution speed, and low rotatory speed). (2,2,6,6-tetramethylpiperidin-1-yl)oxidany (TEMPO) was added to the photoresist at 0.3mW to improve the printability of small features. The photocurable resin was poured into cylindrical glass vials (outer diameter 12 mm) and then sonicated to remove air bubbles.”

Regarding printing with different materials: we have included new printing results in a hydrogel: gelatin Metacryloyl (GelMa) loaded with biological cells. Other material could also have been used. For example, a recent review of materials for TVAM ³.

Here, we have used an optical vortex PSF to exploit the self-healing properties of Laguerre-Gaussian beams in scattering media⁴ (note that other self-healing beams could also be used such as Bessel beams). The generated beam carried an orbital momentum with a topological charge equal to $\ell = 1$. The printed results in Fig 9 show that by just exploiting the PSF properties, we were able to print all details of the expected 3D model without the pre-compensation of the projection pattern necessary for these cell laden constructs⁵.

We added the following figure and text to the manuscript:

“Figure 9 shows confocal microscope images of a printed 3D structure in a cell-laden hydrogel containing 0.5 million cells per mL (see material and methods). It has been shown that self-

healing beams such as Bessel⁵³ or Vortex beams⁵⁴ penetrate deeper in scattering tissue (50% longer penetration depth for skin⁵³) and retain their spatial shape better than conventional Gaussian beams. We believe both effects contribute to the shape fidelity of the fabricated cell-laden 3D construct. In traditional amplitude VAM with Gaussian beams such shape fidelity could not be produced. Here, we have used an optical vortex PSF to exploit the self-healing properties of Laguerre-Gaussian beams in scattering media⁵⁵ (note that other self-healing beams, such as Bessel beams, could also be used). The generated beam carried an orbital momentum with a topological charge equal to $\ell = 1$. The printed results show that by exploiting the PSF properties, all details of the 3D model were printed without any pre-compensation of the projection pattern necessary for these cell-laden constructs¹³. An analysis of the print fidelity was performed using the fluorescent images, we computed the intersection over union (Jaccard Index) of filtered and thresholded images with its 2D models, which shows a remarkable fidelity in this scattering hydrogel (Jaccard Index $\in [0.81, 0.94]$, Supplementary Fig. 12 and Supplementary Movie S6). These results show that holographic volumetric printing can leverage on PSF engineering to improve print fidelity in scattering materials, which are inherent to bioprinting.”

Fig. 9. **a.** Diagram from CAD design illustrating the area from which the z-stack microscopy images were acquired in the bioprinted construct. **b.** Fluorescence (left) and brightfield (right) confocal images showing the multiacinar cavity inside the 3D cellularized construct. **c.** Montage of 2D images from z-stack acquisition. The Brightfield images displaying the sum projection of the 2D slices were added in the Supplementary Information. **d.** Representative brightfield (i) and multi-channel fluorescence (ii, iii) microscopy images showing the maximum intensity projections of 100 slices (i, ii) and one single slice ($n = 66$).

As well as the following text in the manuscript on the hydrogel preparation, printing, postprocessing and imaging:

Hydrogels

Gelatin Metacryloyl (GelMA) was synthesized from porcine gelatin (Sigma) following the protocol of Van De Bulcke et al.⁶¹ Briefly, Type A bloom = 300 porcine gelatin powder (Sigma, G2500) was dissolved at 10% w/v in Phosphate Buffered Saline (PBS) 1x at 50 °C. 25 mL

methacrylic anhydride (Sigma, 760-93-0) was added dropwise and stirred at 50 °C for 3 hours to functionalize the gelatin. The excess unreacted methacrylic anhydride was removed by centrifugation for 15 minutes at 3000 rpm. Furthermore, the material was actively filtered with PBS against 12 times its volume. The solution was lyophilized for 48h and stored at -20 °C, protected from light. 7% w/v GelMA solution was prepared by dissolving lyophilized GelMA powder in PBS containing Lithium phenyl-2,4,6-trimethylbenzoylphosphinate (LAP, Sigma-Aldrich, 900889) at 0.5 mg mL⁻¹, followed by filter sterilization at 40°C. GelMA solutions were stored at 4°C, shielded from light, for up to 2 weeks.

Cell Culture

Human foreskin fibroblasts (HFF-1) were obtained from ATCC and cultured in Dulbecco's Modified Eagle's Medium (DMEM) without phenol red, supplemented with 1% Penicillin-Streptomycin (Gibco), 2% L-glutamine (Gibco), and 15% FBS (Gibco) under standard conditions (37°C, 5% CO₂).

HoloVAM-bioprinting

For HoloVAM bioprinting, a multiacinar cavity structure was fabricated. Fibroblasts were detached, counted, and centrifuged. Cells at a final density of 0.5 million cells mL⁻¹ were resuspended in GelMA with LAP and gently mixed using a 1000 µL pipette tip. 1 mL of the GelMA + LAP + HFF-1 mixture was dispensed into ethanol-sterilized cylindrical glass vials (12 mm diameter) with hermetically sealing caps under sterile conditions in a biosafety cabinet. The vials were placed in water at 2°C to induce gelation of the GelMA and printed within 1 hour.

Post-processing -Cell-laden Hydrogel

Post-printing, the glass vials were heated to 27 °C for 5 minutes in a water bath. Under sterile conditions in a biosafety cabinet, pre-warmed PBS at 37 °C was gently added to the vials, and they were gently agitated to remove uncross-linked GelMA.

Confocal microscopy

Confocal microscopy was used to image the bioprinted constructs which were fixed in formaldehyde 4% v/v in PBS for 15 minutes, then rinsed twice with PBS. Samples were

permeabilized with 0.2% Triton X-100 in PBS for 10 minutes at room temperature, washed three times for 5 minutes each with PBS, and incubated with ((R)-4-Hydroxy-4-methyl-Orn(FITC)⁷)-Phalloidin (1:60, 0.16 nmol mL⁻¹) in PBST + 1% BSA for 30 minutes at room temperature. They were rinsed with PBS three times for 5 minutes each, stained with DAPI in PBS (1:1000) for 5 minutes, washed once with PBS, and imaged using a motorized inverted confocal microscope (Leica SP8) with a 5x NA 0.15 air objective (WD=13.7 mm, HC PL Fluotar, Leica). Images were visualized and processed using ImageJ and pseudo-colored with the colormaps freely provided by Bio Imaging & Optics Core Facility the at EPFL (<https://biop.epfl.ch/Fiji-Update/luts/>).

9. Holographic Volumetric 3D printing can be found in some published papers, which utilize the interference of three holographic beams to create a 3D hologram. This proposed technique belongs to tomographic volumetric additive manufacturing. Holographic Tomographic Volumetric Additive Manufacturing as the title or term may be more accurate.

Reply: We have changed the name of the manuscript to “Holographic Tomographic Volumetric Additive Manufacturing”

Reviewer #2 (Remarks to the Author):

Review of “Holographic Volumetric Additive Manufacturing” by Álvarez-Castaño et al.

High Level Overview

This work introduces a new volumetric 3D printing technique in which holographic light patterns are used to fabricate parts in a liquid photoresin. In contrast to tomographic volumetric additive manufacturing (TVAM) that produces light patterns through amplitude modulation, the method demonstrate here uses phase modulation and the Lee hologram method for light pattern creation. As such, the authors call this method holographic volumetric additive manufacturing (HoloVAM) and show remarkable print quality of a 3D Benchy test print. In particular, they demonstrate millimetric scale printing with micron-scale feature sizes. The increase in print resolution can be attributed to the use of self-healing Bessel beams in conjunction with time-multiplexing to transmit patterns at different planes within the print volume. Furthermore, it is shown that HoloVAM is 10-fold more light efficient than TVAM since all pixels on the spatial light modulator can contribute to light pattern creation.

Validity

Interpretation of the data was well done, and conclusions justified and reasonable. My main suggestion is to incorporate the microCT data into the analysis to further improve the quality of the manuscript and support their claims.

Reply: We thank the reviewer for the suggestion. We have included a quantitative analysis using a microCT to perform a Jaccard similarity measurement on a 3D construct (Benchyboat) in acrylate material. We have also conducted a quantitative analysis using confocal microscopy to measure the bioprinted construct in hydrogels. Additionally, we use the microCT to measure the stria reduction analysis. See response #2 and a figure was added to Supplementary Information S12.

The analysis has been included in the results section.

Significance

The addition of holographic methods to tomographic volumetric printing will open new frontiers in this printing modality. As such, I expect this paper to be of high impact and play a transformative role in the additive manufacturing landscape.

Reply: We thank the reviewer for appreciating the novelty of our approach and recognizing the advantages of our method.

High-Level Comments

1. The paper provides introduces a new volumetric 3D printing approach which improves upon TVAM via increasing the printable volume (Bessel beams + multiplane projection) and increasing light efficiency. The authors also state that phase-encoding uses a *simpler optical setup* (than TVAM). While I agree that in its simplest form HoloVAM does use a simpler optical configuration, the authors motivate the strength of their approach using a more complicated system. I.e. using Fresnel lenses to achieve multiplane projection and an Axicon to achieve Bessel beams, both of which adds complexity beyond that of TVAM. Prior literature has shown TVAM capable of making parts with comparable resolution to HoloVAM without the need for either Fresnel or Axicon components. I recommend that the authors either remove or rephrase this statement.

Reply: The flexibility of the HoloVAM phase modulation allows the Axicon and Fresnel lenses to be encoded digitally together with the phase pattern that corresponds to the holographic projection in the SLM. Therefore, the Axicon and Fresnel are not additional components in the projection system.

We clarified this point in the manuscript by adding:

“Second, phase-encoding allows a modification of the point spread function (PSF) to be encoded within the same hologram, allowing 3D digital control of the light beam, for example, by increasing the depth of focus using an Axicon phase to generate a Bessel beam, or by multi-plane projections of the same pattern by adding phases of Fresnel lenses, effectively creating a low-divergence projection.”

2. An index-matching bath surrounds the print volume in HoloVAM. In TVAM, it has been demonstrated that this bath can be removed via a suitable remapping of the Radon coordinates or explicitly using optical ray tracing. Given this information, could HoloVAM be expanded to not use an index-matching bath to further simplify the system? Further, could this method be expanded to a non-telecentric (e.g. cone beam) configuration?

Reply: Indeed. Thank you for pointing this out. Any correction performed in conventional TVAM could be applied to HoloVAM. We did not add this extra step in our study in order not to add complications. In addition, one of the advantages of the phase encoding is the possibility to add a phase compensation of a characterized distortion, such as the optical aberration of the system or, more specifically, the wavefront distortion generated by the slight difference between the index of refraction of the vial walls and the photoresin. Furthermore, the digital beam control of the phase encoding allows the system to be extended to a non-telecentric configuration. A cone-beam configuration could be an interesting option, however we have not investigated this in this study. There remains many interesting further developments to take full advantage of phase encoding in TVAM.

Detailed Comments

1. On Line 95, “...reach negative features (holes) of 50 μm reliably”. I do not agree with the “reliable” part of this statement. A series of 3D Benchy parts are shown in Figure 6 for different tile number and PSF. In particular, feature sizes for the chimney, cargo box, and anchor hole are shown, and vary widely depending on PSF and tile number. Of the four parts shown, only Fig. 6 c,d) show negative features less than 50 μm and furthermore, are much smaller than the target size of 97 μm . Has a repeatability study been performed to assess print-to-print variation?

Reply: It is important to clarify that the target size of 97 μm is for a specific feature of the benchy boat. However, for a given target size, the printed part could show a different printed size due to the dependence on the optical dose received by the resin. If the dose is low, then the printed size could be smaller than expected, and if the dose received is higher, the

expected size is larger until a saturation is reached. Furthermore, if the target object has small features, diffusion plays a role where the small features are not printed or are difficult to print while the large features reach the correct dose before. Here, the additional print time required for small features to be printed causes over-polymerization on certain features.

We added the following figure and text to the supplemental

S13. Dose test

To quantify the printed object fidelity, multiple cylinders with a hole in the middle were printed, by varying the printing time. The cylinder height and perimeter of the inner hole were measured. From the results in the plots below, we can see that there can be small variations of features (height and perimeter) for printing time in the range of milliseconds. However, close to 100% fidelity could be obtained for a print time of 37 seconds.

Supplementary Fig 11. Height a) and perimeter b) of a printed cylinder containing a hole in its middle as a function of printing time.

2. Line 406: MicroCT imaging. Based on your MicroCT imaging, can you quantify the printed part accuracy beyond that of the microscope images? E.g. 3D Jaccard similarity, signed-distance field, etc.

Reply:

From the microCT scans we measure the Jaccard similarity index or the intersection over the union between the printed part and the MicroCT, where Jaccard = 1 means perfect agreement. Supplementary Fig. 12. Illustrates a representative 2D slice from the microCT with its corresponding slice from the 3D model for the 4 printed parts shown in Fig.7c. Here we can see a print fidelity up to 0.86 for the Benchy boat printed with an acrylate-based resin printed with a Flat-top, Supplementary Fig. 12 b). The ~3% difference from the other prints is related

to the photochemistry of the photoresin, where TEMPO was added to overcome the oxygen diffusion, allowing us to print more features of the object.

In addition, we evaluate the fidelity of the construct printed in the cell-laden hydrogel. Despite evaluating the fidelity of constructs printed in hydrogel, where they are soft and could deform, we have performed a Jaccard similarity index to evaluate the fidelity using the fluorescence microscopy images of stained cell nuclei. Supplementary Fig.12 c), shows a comparison of representative cross-sections of confocal microscope images of the hydrogel construct (complex acinus) with the 3D model slices.

We have added to Supplementary Information S14 the following image:

Supplementary Figure 12. **a**) Comparison of a representative cross-section of the microCT image of a acrylate printed part (Benchy boats from Figure 7c) with a 2D slice from the 3D model. **b**) Measurements of print fidelity using the Jaccard similarity index as the intersection over the union of model and print, where Jaccard = 1 means perfect agreement. The measurements were performed for the 4 samples from Figure a. **c**) Comparison of representative cross-sections of confocal microscope images of the hydrogel construct (complex acinus) with the 3D model.

3. Line 340. Is the rotation rate of the vial (30 deg/s) chosen based on a limitation, such as the framerate needed for multiplane projection or non-laminar fluid flow in the vial?

Reply : We use a DMD from Vialux DLP7000 as a display to project the binary phase holograms. The DLP7000's switching rate is 22'727 Hz for 1-bit images, 1091 Hz for 6-bit

images, and 290 Hz for 8-bit images. Thanks to the Lee hologram methods, which allow us to use the DMD as a fast phase modulator with binary holograms, we can display 1-bit images. For the experiments, the display rate was fixed at 1.6 kHz, which allows to accumulate a sufficient dose per angle given the available optical power, and a printing time in the order of a minute. With this setting, the system displays from 1 up to 32 holograms per angle, with an angular resolution of 0.6° and an exposure time of 20 ms per angle. At $30^\circ/\text{s}$ we are not observing any flow that affects the projected patterns.

We added the following clarification on the rotation speed of the vial and the projection rate in the manuscript:

“The collimated beam illuminates a DMD (Vialux DLP7000: $L \times W = 768 \times 1024$ micro-mirrors and pixel pitch = $13.6 \mu\text{m}$) placed in a Fourier configuration and aligned to meet the blazed grating criteria of efficiency. After the Fourier lens $L1$ ($f_1 = 150 \text{ mm}$), a spatial filter (SF) was added to filter out the zero-order and keep the -1 order. $L2$ ($f_2 = 150 \text{ mm}$) and $L3$ ($f_3 = 200 \text{ mm}$) form a 4-f system conjugating the Fourier plane and rescaling the holograms on the sample plane. The DMD was synchronized with the rotary stage (Zaber, RSW60C-E03T7-KX13A) using a Data Acquisition Card from National Instruments (X series DAQ, PCIe-6321) at a typical framerate of 1600 Hz. The rotatory stage holding the sample vial is set to rotate at a constant speed ($30^\circ/\text{s}$) while a sequence of holographic projections is displayed every $\Delta\theta = 0.6^\circ$ synchronously on the DMD. The choice of 30 degree/s is a typical value for VAM printers. However, rotation speed could be increased as the projection rate of the DMD is 22'700 images (holograms) per second.”

4. Discussion. Another benefit of binary-pattern projection is increased projection frame rate. Binary images take up less memory than typical gray-scale images in TVAM (e.g. 8-bit).

Reply: Thank you for the suggestion. We had added a discussion about 1-bit images the conclusion and discussion section

“Moreover, different from conventional TVAM, the binary phase patterns are 1-bit images, which allows us to use the maximum frame rate of the DMD $\sim 22 \text{ kHz}$, whereas TVAM uses 8-bit images, which limits the rate of the DMD at 290 Hz. The high frame rate of 1-bit images and the print setting of our system enables us to multiplex on time up to 36 holograms per angle.”

5. How similar is the refractive index (RI) of the vegetable oil to your photocurable resin ($n \sim 1.47$)? Based on the stated number of significant digits I assume your RI is between [1.465, 1.474] depending on method of rounding. If the RI wasn't perfectly matched, I would expect a magnification (or demagnification) of your projected field. Do you observe any such

effect? At such small feature scales (10s of μm), would a RI error of ± 0.005 play a significant role on your print fidelity?

Reply: Good point. We have measured the refractive index of the resin and the vegetable oil, which corresponds to $n_{resin} = 1.4833$ and $n_{oil} = 1.4763$ respectively. This results in a mismatch error of error of $n\Delta = 0.007$. To theoretically calculate the magnification of the system, we have simulated our optical system in ZEMAX from the last Fourier lens to the resin container, taking into account the NA of our system, the lens focus, different surface geometries, and their refractive index. From the simulation, we have calculated a magnification of 1.004x. We have considered this magnification when computing the projected patterns.

In addition, we have taken images of a specific projected pattern through all three media (air, oil, and resin) using the inspection camera 2 to measure the image size and quantify experimentally the magnification. Here, a 3D model of a cylinder with a hole has been used to compare the sizes of one holographic projection of the angle 0° , Supplementary Fig. 10. The size of the inner diameter was measured in the three different media. The size on air is $715.36 \mu\text{m}$ and in resin after passing through the three media and surfaces is $719.20 \mu\text{m}$. Therefore, the magnification is 1.005x.

Supplementary Fig. 10.2. a. 3D model of a cylinder with a hole. b-d. Reconstruction of the holographic projection corresponding to the 0° angle in air, oil, and resin respectively.

It is important to note that HoloVAM requires an alignment process each time we change the photoresin material and index matching bath because the axes of rotation should coincide with the Fourier lens focus. The alignment protocol uses the inspection system associated with camera 1, where the magnification is 1x.

We have included the following text to the supplementary Information:

“S. 11 Index Matching Bath: Magnification Analysis

To determine the possible magnification generated for a refractive index mismatch between the resin and the photoresin contained in the sample holder. We modeled the system using Zemax. We measure the refractive index of the vegetable oil and photoresin using a refractometer, which corresponds to $n_{resin} = 1.4833$ and $n_{oil} = 1.4763$ respectively. This results in a mismatch error of error of $n\Delta = 0.007$. We used ray tracing in the image plane (Fourier lens focus) and obtained spot diagrams (maps of the position of the ray intersecting the image plane) for three different points. In the first case, ray tracing in air, we could see that the reconstruction points are infinitesimally small, Supplementary Fig. 10.1 a). We also modeled the complete system, where the cylindrical glass resin container (diameter = 12 mm) is immersed on a glass cuvette with vegetable oil. The different surfaces, real dimensions and different refractive indices are considered. The spot diagrams obtained show a small magnification of the spots. However, the system does not introduce any significant magnification and provides magnification information of 1.004x, which is almost negligible.”

Supplementary Fig 10.1. Spot diagrams of the ray tracing simulations in Zemax. **a** Results of the spot diagram produced if the reconstruction image occurs in air. **b** Results of the spot diagram produced if the reconstruction image occurs after the ray propagated through all the surfaces and different mediums (air, oil, and resin). Scale bars (vertically located on the left side) are 0.02 μm and 0.10 μm , respectively.

The simulated results are compared with the experimental results. A single holographic projection from the 3D model of a hole cylinder (Supplementary Fig. 10.2. a) is calculated from a specific angle 0° . Images of the reconstructed intensity in the Fourier plane in the three different media (air, oil, and resin) are collected using the inspection camera 2. The size of the inner diameter was measured in the three different media. The size in air and in the resin is 715.36 μm and 719.20 μm respectively. The magnification is therefore 1.005x.

Supplementary Fig. 10.2. a. 3D model of a cylinder with a hole. b-d. Reconstruction of the holographic projection corresponding to the 0° angle in air, oil, and resin respectively.

Sentence-level clarity improvements

1. Line 15. Replace “into a resin” to “into a photo-curable resin” or “into a photoresin”.

Reply: We replaced the term «resin» in the manuscript by “photoresin”.

2. Line 183. “There is a clear trend of decreasing the error of the projected hologram while the PSNR increases standing a large speckle noise reduction by performing time multiplexing.” Please consider rephrasing as it is currently confusing to read. I understand this statement as: “For increasing number of tiles, MSE and speckle noise decreases while PSNR increases, with further improvement with time-multiplexing.”

Reply: We have rephrased the sentence in the manuscript:

“there is a clear trend towards increasing the PSNR and decreasing MSE when the number of tiles increases. This is due to the speckle noise reduction and smaller errors of the holographic projections with increasing number of tiles, with additional enhancement through time multiplexing.”

3. Line 332. “L X L = 768 X 1024 ...”. Do you mean “L X W = 768 X 1024” ?

Reply: The change has been included.

Formatting improvements

Body

1. General comment about terminology consistency. The terms “resin” and “photoresin” are used interchangeable. Please be consistent with your usage and choose a single term.

Reply: We have made the necessary changes.

2. Line 41. In this paragraph tomographic volumetric additive manufacturing is given the acronym TVAM, but yet later in the paragraph is still explicitly written out.

Reply: We have made the correction in the manuscript:

“Unlike most other additive manufacturing methods, TVAM is layer-less, meaning that it does not fabricate objects by solidifying one voxel, line, or layer at a time.”

3. Line 73. “We term this novel technique HOLOVAM”. Font size appears larger than body.

Reply: We have corrected the font size and the font type.

4. Line 147. Add space between “Supplementary Fig”.

Reply: The changes have been incorporated.

5. Line 223. Replace “With the aim of use a light engine base on” with “With the aim of using a light engine based on”.

Reply: We have included the change.

6. Line 228. Replace “We exploit the PSF engineering reducing” with “We exploit the PSF engineering by reducing”

Reply: We have included the change

7. Line 231. Replace “non-diffractive beams, Bessel beam produce a” with “non-diffractive source, such as a Bessel beam, produces a”

Reply: We have edited the sentence as follows:

“second, a non-diffractive source, such as a Bessel beam, produces a beam with low etendue”

8. Line 239: Add space between “10mm”.

Reply: The correction was made.

9. Line 267: Add space between “Ntwere”

Reply: The correction was made.

Figures

4. Figure 2b: Histogram of pixel gray level has x-axis range from [0,1]. However, on Line 123 gray level is defined as ranging from [0,255]. Either change Line 123 to match Figure

Clarify the gray level for tomographic projections.

Reply: For tomographic amplitude projections 8-bit images are used, therefore the Gray level range is defined in a range between [0,255]. However, the histogram is plotted with a normalized Gray level, so the range is [0,1]. To clarify this point we have changed the equation, the equation description, and the x-axis label in Fig 2b. The changes are as follows:

$$\eta_{patt} = \frac{\sum_{g=0}^{g_{max}} n_g \frac{g}{g_{max}}}{N_{pixels}} * \eta_{DMD} \quad (1.0)$$

“where n_g is the number of pixels in the image that have a gray level equal to g , and g_{max} is the maximum gray level value, where 8-bit images are used for amplitude tomographic projections, therefore $g_{max} = 255$. N_{pixels} is the number of pixels of the image, and η_{DMD} is the pixel reflectivity of the DMD at the operating wavelength of 405 nm.”

The Fig. 2. has been changed as follows:

2, or vice-versa.

5. Figure 2c: Does the phase pattern include the PSF?

Reply: Yes, it does. The phase projections encode the phase retrieval from the GS algorithm, which allows the intensity reconstruction of the required tomographic projection and must also include the PSF modification required for our speckle-noise reduction technique (HoloTile).

To clarify, we have rephrased the caption as:

“c) (Left) Corresponding phase pattern (hologram) generated by the HoloTile technique, encoded as a Lee hologram to be able to use the DMD as a fast phase modulator. (Right) Reconstructed projection of the Benchy boat from a coherent projection (hologram).”

6. Figure 4c, caption: Replace “Bessel PDF” with “Bessel PSF”

Reply: The correction was made.

References

1. Please make sure references adhere to Nature formatting guidelines (<https://www.nature.com/nature/for-authors/formatting-guide>). Several references are listed with more than 5 authors (e.g. References 8,9) which should be replaced with [First Author] et al.

Reply: We have changed the citation style to the Nature format in the main text and in the Supplementary Information.

Reviewer #3 (Remarks to the Author):

The manuscript presents what the authors call holographic volumetric additive manufacturing in which they explore phase rather than amplitude approach to Tomographic Volumetric Additive Manufacturing (TVAM). They use a binary amplitude spatial light modulator in the form of a micromirror array (DMD) to create a phase hologram with the Lee method that is then optically reconstructed in the resin. As written, the manuscript is at best confusing, which leaves me recommending rejection or possibly major revision because it is difficult to evaluate the authors' approach and results. Some comments and examples are included below.

Questions/comments:

- Fig. 1 claims to show the optical set up of the system. But later in the text Fig. S1 is referred to as the actual optical set up. Unfortunately, the two are quite different and cause significant questions. For example, Fig. 1 does not show diffraction order spatial filtering, which is crucial in Lee hologram method to convert binary amplitude modulation from the DMD to phase modulation. However, diffraction order spatial filtering is present in Fig. S1. Why the conflicting figures of the system?

Reply: It is true that in our experiment, we have used a spatial filter in the Fourier plane of relaying lenses which makes the system bulkier than Fig 1 S1. This is a commonly used method in the Lee method. The objective here was not to minimize the projection system.

There is actually a large body of work currently in augmented reality glasses to minimize holographic projection systems. For example in Wetzstein et al “Holographic Glasses”⁹, there is no spatial filter. The diffraction orders and zero orders are mixed to form the holographic reconstruction. Hence more sophisticated algorithm could be used to make the setup more compact in the future. Here, the objective of this manuscript was to demonstrate the flexibility of phase encoding for tomographic VAM.

- Relatedly, the Fig. S1 caption says, "A single lens allows the reconstruction of the projected holograms into the rotatory resin container...", but Fig. S1 shows that there are actually several lenses.

Reply: To avoid confusion we restated the caption and description of Fig. S1. The caption had changed as follows:

“A Fourier lens L1 reconstructs the hologram at the Fourier plane, which is then imaged into the rotatory resin container. Using lenses L2 and L3 we conjugate the Fourier plane (green dashed line illustrates the conjugated planes).”

The Supplementary Fig.1. change is as follows:

Supplementary Fig. 1. Experimental setup of the Tomographic Volumetric 3D printing by phase encoding (HoloVAM). A Fourier lens L1 reconstructs the hologram at the Fourier plane, which is then imaged into the rotatory resin container. Using lenses L2 and L3 we conjugate the Fourier plane (green dashed line illustrates the conjugated planes).

It is important to note that the use of the 4F system composed of lenses L2 and L3 is not absolutely necessary. By choosing an appropriate carrier for the Lee hologram coding, the zero order could be made far enough away from the intensity reconstruction, then the rotary stage and the print sample could be placed directly in the Fourier plane.

In addition, we modified Fig 1. We added the description of the Fourier lens on the figure and caption.

Fig. 1 Optical configuration for holographic volumetric additive manufacturing. A single-frequency laser diode at 405 nm is collimated and expanded to fit the active area of a DMD. A Fourier lens reconstructs the hologram at its Fourier plane which is located within the rotatory photoresin container (more details in fig 1 supplemental). The holographic projections are displayed synchronously with the rotation stage. Two inspection cameras are used to monitor the holographic pattern reconstruction camera, and the polymerization process camera 2 (See Supplementary Fig. 1).

- The authors state that the laser diode is single mode, which I assume refers to its spectral properties. What is the actual spectral width of the source and therefore its coherence length? How does the coherence length compare to the critical lengths in the hologram reconstruction and is the necessary degree of coherence maintained?

Reply: The term single mode refers to the spatial mode in the text. The spectral characteristics of the laser diode source is such that the angular dispersion due to the blazed grating of the spatial light modulator does not introduce aberrations of the PSF. Coherence length is 596 mm.

Such a narrow spectral bandwidth is not a strict requirement if other coding schemes are used. Broad band single mode sources such as SLEDs would be an interesting choice for further study. Thank you for the bringing it up.

- Eq. 1 indicates that the authors use grayscale images on the DMD. Such images are generated through temporal modulation of the fraction of the time a given pixel is "on" during a frame time. What are the effects in the image region in the resin container when at any given instant in time only some of the pixels are on, and in the next instant some of them have turned off while other pixels have turned on in order to hit the graylevel for each of the pixels, which is determined automatically in the DMD drive chip? The authors analysis implicitly asserts that this is the same situation as having the light amplitude from each pixel be continuously on at the proper graylevel for the entire frame time. This seems problematic, but at a minimum should be justified and confirmed.

Reply: Lee holograms are binary holograms meaning the pixels are either "on" or "off" on the DMD for any given frame. When projected, the holograms can give a varying intensity image (hence gray scale) in the vial.

A different hologram corresponds to a different binary pattern. With a given binary pattern (a frame) on the DMD, there is no temporal modulation. The frame rate of the holograms can be thus up to the maximum (binary) frame rate of the DMD (22kHz). (see answer question 3 reviewer 2).

In contrast, amplitude coding requires time modulation of each pixel to code the grayscale pattern.

We hope this clarifies the misunderstanding.

- What's the minimum exposure any given small region of resin receives? If it is zero, how is this explained, and how does this compare to amplitude TVAM?

Reply: The minimum exposure could be zero. Zero intensity can be generated with a CGH via destructive interference. The minimum exposure of any given region depends on the 3D printed object. For example, in Fig. 5, the center of the extruded gear has near zero exposure. The holograms are computed to yield the wanted projection patterns. Fig. 5 c shows simulation and experiment, both having very low exposure in the center.

Unlike other light-based AM methods, in VAM and HVAM the pattern projections change at every angle during a turn. This creates intensity fluctuations over time at each voxel in the print volume. These fluctuations cause diffusion effects that make it difficult to print small features because the energy received in time is not enough to deplete the oxygen necessary to polymerize the voxel. These effects are negligible in techniques such as DLP. Intensity variations are pattern-dependent and therefore object-dependent. The following figure illustrates the intensity variation for each voxel. Some voxels have intensities equal to zero.

- a) Tomographic projection Benchy angle $\theta = 0^\circ$. b) intensity fluctuation of voxel (b) during the 360°. c) intensity fluctuation of the light intensity of the analyzed pixel (c) during the 360°.

Since the tomographic projections are the target intensities for or phase retrieval algorithm, we expect that the dose received for each voxel will be the same using amplitude projections as phase projections. Phase patterns will provide more light efficiency, better resolution, and high flexibility of the system, allowing us to control the light axially, change the PSF, and it will allow us to correct distortions in the system.

- For the Lee method implementation, what is the trade-off the authors have made between resolution and fidelity of the generated phase pattern based on choice of carrier spatial frequency, spatial sampling in terms of how many pixels per macro-pixel, and the size of the iris used for spatial frequency filtering for the -1 order? Given the limited number of pixels on the DMD, what does this imply about the size and resolution of what can be fabricated in the resin?

Reply: The resolution of our system is currently limited by the speckle noise reduction technique (Holotile). The process of tiling the holograms reduces the speckle noise because the reconstruction points are separated in the Fourier to avoid the overlapping between adjacent frequency components¹⁰. Therefore, we have defined the parameter “minimal feature size”, which is the “output pixel” in the reconstruction plane and is highly dependent on the

number of tiles used. This parameter is related to the size of the full hologram ($L \times W$), the number of tiles (N_t), the pixel size ℓ_{px} , the wavelength λ , and the focal length of the Fourier lens. The minimal feature size is the minimum feature that we can print and depends on the number of tiles. This parameter does not depend on the Lee hologram method, the same definition of minimum feature size needs to be used if instead, we use a phase-only SLM with HoloTile for speckle noise reduction.

In addition, the size limitation is dependent on the light output and has been addressed in our response to point 1) of Reviewer #1.

The resolution is discussed in Fig. 3 and in the text accompanying the figure.

“the projection’s minimal feature size is the distance between the grid in the reconstruction plane (Fig. 3 f). We improve the speckle noise at the cost of an increased feature size.”

Fig 3. Hologram tiling reduces speckle noise. **a** Schematic of the HoloTile hologram process, where a sub-hologram is spatially tiled and convolved with a Flat-top PSF. **b** Experimental reconstruction of the tiled holograms generated of the letter

"A" for a different number of tiles N_t . The experiments were performed using a liquid crystal SLM (See Supplementary Information note S2, S-Figure 3). Comparison of the measured **c** mean square error (MSE) and **d** peak signal to noise ratio (PSNR) for the different tiled holograms. **e** Reconstruction analysis considering the contrast measurements as $(I_{max} - I_{min}/I_{max} + I_{min})$. **f** Minimal projected feature size measured for different tiled holograms using the DMD. Experimental reconstruction using the DMD are shown in Supplementary Information S2.1.

Reviewer #4 (Remarks to the Author):

References

1. Webber, D. *et al.* Micro-optics fabrication using blurred tomography. *Optica* **11**, 665 (2024).
2. Rackson, C. M. *et al.* Latent image volumetric additive manufacturing. *Opt. Lett.* **47**, 1279–1282 (2022).
3. Madrid-Wolff, J. *et al.* A review of materials used in tomographic volumetric additive manufacturing. *MRS Communications* **13**, 764–785 (2023).
4. Daria, V. R., Palima, D. Z. & Glückstad, J. Optical twists in phase and amplitude. *Opt. Express* **19**, 476 (2011).
5. Madrid-Wolff, J., Boniface, A., Loterie, D., Delrot, P. & Moser, C. Controlling Light in Scattering Materials for Volumetric Additive Manufacturing. *Advanced Science* **9**, 2105144 (2022).
6. Fahrbach, F. O., Simon, P. & Rohrbach, A. Microscopy with self-reconstructing beams. *Nature Photon* **4**, 780–785 (2010).
7. Bouchal, Z. Resistance of nondiffracting vortex beam against amplitude and phase perturbations. *Optics Communications* **210**, 155–164 (2002).
8. Van Den Bulcke, A. I. *et al.* Structural and Rheological Properties of Methacrylamide Modified Gelatin Hydrogels. *Biomacromolecules* **1**, 31–38 (2000).

9. Kim, J. *et al.* Holographic Glasses for Virtual Reality. in *Special Interest Group on Computer Graphics and Interactive Techniques Conference Proceedings* 1–9 (ACM, Vancouver BC Canada, 2022). doi:10.1145/3528233.3530739.
10. Madsen, A. G. & Glückstad, J. HoloTile: Rapid and speckle-suppressed digital holography by matched sub-hologram tiling and point spread function shaping. *Optics Communications* **525**, 128876 (2022).

Rebuttal to “Holographic Tomographic Volumetric Additive Manufacturing”: Addressing Reviewer Comment #1

We thank the editor for the handling of our manuscript and the 4 reviewers for re-assessing our answers. We hereby address the remarks from reviewer #1.

REVIEWER COMMENTS

Reviewer #1 (Remarks to the Author):

Thanks for the revision. However, I still don't understand why holographic projection is used to replace simple image projection. Lee's method was used to smoothen the holographic projection, but still, it is not comparable to the image projection quality. This can be obviously noticed from the printed samples, whose surface is very rough and even the shapes are significantly deformed. I was not convinced by the novelty and quality of this paper as NC journal.

Reply:

The manuscript investigates the modality of holographic projection instead of pure amplitude projection. In this work, we have changed the light engine from amplitude to phase modulation. Why is it used? Holographic projection provides a mean to control the depth of projection and hence gives more possibilities than amplitude coding. The manuscript investigates this, including a 10X improvement in light efficiency demonstrated experimentally with an optimal light source etendue (diffraction limited) to achieve the best resolution possible theoretically. Of course, additional improvements are needed to obtain an optimal projection quality. We cited recent work by Wetzstein et al , e.g. that use neural nets to further remove speckles in the projection. Lee holograms are not optimal because of loss of fidelity (due to binary amplitude to phase coding) but it was necessary given the 405 nm wavelength used for printing. There are upcoming pure phase SLM pistons that will further increase projection efficiency and fidelity. We have also experimentally demonstrated the uniqueness of holographic projection by forming self-healing beams, useful for printing in scattering media without pre-compensating the amplitude targets (demonstrated printing in a cell laden hydrogel). This is not possible using pure amplitude projection. In summary, phase patterns provides vastly more light efficiency, better resolution

(single mode spatial light sources), and high flexibility of the system, allowing us to control the light axially, change the PSF which will allow to correct distortions in the system.

Regarding the Lee hologram comment: the Lee method is not used to smooth the projection. It is used to transform the binary amplitude DMD as a fast phase modulator because the proposed multi focus method requires fast modulation (kHz) and commercially available liquid crystal-based phase-only SLMs are slow (and not stable at 405 nm). It is true that the sample have striations in the order of 20um. However, we showed that the surface quality can be improved experimentally thanks to PSF engineering as shown in the article Fig. 8. We note that the amplitude coding method also produces striations of 100 um in size. Striation is inherent to the tomographic printing technique and a combination of material development (less shrinkage, index change upon irradiation) and optical methods remain an active research area.

We have included changes to the abstract and introduction:

Abstract:

*Several 3D light-based printing technologies have been developed that rely on the photopolymerization of liquid resins. A recent method, so-called Tomographic Volumetric Additive Manufacturing, allows the fabrication of microscale objects within tens of seconds without the need for support structures. This method works by projecting intensity patterns, computed via a reverse tomography algorithm, into a photocurable resin from different angles to produce a desired 3D shape when the resin reaches the polymerization threshold. **Printing using incoherent light patterning has been previously demonstrated. In this work, we show that a light engine with holographic phase modulation unlocks new potential for volumetric printing. The light projection efficiency is improved by at least a factor 20 over amplitude coding with diffraction-limited resolution and its flexibility allows precise light control across the entire printing volume. We show that computer-generated holograms implemented with tiled holograms and point-spread-function shaping mitigates the speckle noise which enables the fabrication of millimetric 3D objects exhibiting negative features of 31 μm in less than a minute with a 40 mW light source in acrylates and scattering materials, such as soft cell-laden hydrogels, with a concentration of 0.5 million cells per mL.***

Introduction:

*.....In this work, **we propose a light engine that harnesses the phase properties of the light beam. We term this technique HoloVAM. Phase encoding of tomographic projections offers multiple advantages over amplitude modulation. First, phase-encoding improves light efficiency, as all pixels of the display can contribute to the projected intensity pattern. Since tomographic projection patterns typically exhibit high spatial frequency information, most pixels are dark. Amplitude encoding is therefore highly inefficient: typically, less than 1% of the incident light***

reaches the vial. Second, phase-encoding allows a modification of the point spread function (PSF) to be encoded within the same hologram, allowing 3D digital control of the light beam, for example, by increasing the depth of focus using an Axicon phase to generate a Bessel beam, or by multi-plane projections of the same pattern by adding phases of Fresnel lenses, effectively creating a low-divergence projection. A self-healing beam, generated using a Helical Phase Plate (HPP) to produce an Optical Vortex (OV), enables printing within scattering materials. Herein, we use a projection system that converts a two-dimensional phase modulator to a two-dimensional intensity pattern in the Fourier plane. All pixels on the modulator contribute, by interference, to all pixels on the image plane.....

Reviewer #2 (Remarks to the Author):

We appreciate the author's thorough response to all questions raised. As such, we recommend the manuscript for publication.

Reviewer #3 (Remarks to the Author):

The authors have adequately addressed the concerns raised in my original review.

Reviewer #4 (Remarks to the Author):

We thank reviewers # 2,3 and 4 for their comments.